



# Evaluation of MOPITT version 7 joint TIR-NIR $X_{CO}$ retrievals with TCCON

Jacob K. Hedelius[1], Tai-Long He[1], Dylan B. A. Jones[1], Rebecca R. Buchholz[2], Martine De Mazière[3],
Nicholas M. Deutscher[4], Manvendra K. Dubey[5], Dietrich G. Feist[6,7], David W. T. Griffith[4], Frank Hase[8],
Laura T. Iraci[9], Pascal Jeseck[10], Matthäus Kiel[11], Rigel Kivi[12], Cheng Liu[13], Isamu Morino[14],
Justus Notholt[15], Young-Suk Oh[16], Hirofumi Ohyama[14], David F. Pollard[17], Markus Rettinger[18],
Sébastien Roche[1], Coleen M. Roehl[11], Matthias Schneider[8], Kei Shiomi[19], Kimberly Strong[1],
Ralf Sussmann[18], Colm Sweeney[20,21], Yao Té[10], Osamu Uchino[14], Voltaire A. Velazco[4,22], Wei Wang[23],
Thorsten Warneke[15], Paul O. Wennberg[11,24], Helen M. Worden[2], and Debra Wunch[1]

[1]Department of Physics, University of Toronto, Toronto, Canada
[2]Atmospheric Chemistry Observations & Modeling, National Center for Atmospheric Research, Boulder, Colorado, USA
[3]Royal Belgian Institute for Space Aeronomy (BIRA-IASB), Brussels, 1180, Belgium
[4]Centre for Atmospheric Chemistry, School of Earth, Atmospheric and Life Sciences, University of Wollongong, Northfields
Ave. Wollongong NSW 2522 Australia
[5]Los Alamos National Laboratory, Earth and Environmental Sciences, Los Alamos, NM, USA
[6]Deutsches Zentrum für Luft- und Raumfahrt, Institut für Physik der Atmosphäre, Oberpfaffenhofen, Germany
[7]Max Planck Institute for Biogeochemistry, Jena, Germany
[8]Institute of Meteorology and Climate Research (IMK-ASF), Karlsruhe Institute of Technology, Karlsruhe, Germany
[9]NASA Ames Research Center, Mountain View, CA, USA
[10]LERMA-IPSL, Sorbonne Université, CNRS, Observatoire de Paris, PSL Université, 75005, Paris, France
[11]Division of Geological and Planetary Sciences, California Institute of Technology, Pasadena, CA, USA
[12]Finnish Meteorological Institute, Sodankylä, Finland
[13]University of Science and Technology of China, Hefei, 230026, China
[14]National Institute for Environmental Studies (NIES), Tsukuba, Japan
[15]University of Bremen, Bremen, Germany
[16]National Institute of Meteorological Sciences 33, Seohobuk-ro, Seogwipo-si, Jeju-do 63568, Republic of Korea
[17]National Institute of Water and Atmospheric Research, Lauder, New Zealand
[18]Karlsruhe Institute of Technology (KIT), Institute of Meteorology and Climate Research (IMK-IFU),
Garmisch-Partenkirchen, Germany
[19]Japan Aerospace Exploration Agency, Tsukuba, Japan
[20]Cooperative Institutes for Research in Environmental Sciences, University of Colorado, Boulder, CO, USA
[21]NOAA Earth System Research Laboratory, Boulder, CO, USA
[22]Oscar M. Lopez Center for Climate Change Adaptation and Disaster Risk Management Foundation, Inc., Philippines.
[23]Key Laboratory of Environmental Optics and Technology, Anhui Institute of Optics and Fine Mechanics, Chinese Academy
of Sciences, Hefei, China
[24]Division of Engineering and Applied Science, California Institute of Technology, Pasadena, California, USA

**Correspondence:** Jacob K. Hedelius (jacob.hedelius@atmosp.physics.utoronto.ca)

**Abstract.** Observations of carbon monoxide (CO) from the Measurements Of Pollution In The Troposphere (MOPITT) instrument onboard the Terra spacecraft were expected to have an accuracy of 10 % prior to launch in 1999. Here we evaluate MOPITT version 7 joint TIR-NIR (V7J) accuracy and precision, and suggest ways to further improve the accuracy of the obser-



vations. We take five steps involving filtering or bias corrections to reduce scatter and bias in the data relative to other MOPITT soundings, and ground based measurements. 1) We apply a preliminary filtering scheme in which measurements over snow and ice are removed. 2) We find a systematic pairwise bias among the four MOPITT along-track detectors (pixels) on the order of 3–4 ppb with a small temporal trend, which we remove on a global scale using a temporally trended bias correction. 3) Using a

small region approximation (SRA) a new filtering scheme is developed and applied based on additional quality indicators such as signal-to-noise. After applying these new filters, the root mean squared error computed using the local median from the SRA over 16 years of global observations decreases from 3.84 ppb to 2.55 ppb. 4) We also use the SRA to find variability in MOPITT retrieval anomalies that relates to retrieval parameters. We apply a bias correction to one parameter from this analysis. 5) After applying the previous bias corrections and filtering, we compare the MOPITT results with the GGG2014 ground-based Total

Carbon Column Observing Network (TCCON) observations to obtain an overall global bias correction. These comparisons show that MOPITT V7J is biased high by about 6–8 %, which is similar to past studies using independent validation datasets on V6J. When using TCCON spectrometric column retrievals without the standard airmass correction or scaling to aircraft (WMO scale), the ground and satellite based observations overall agree to better than 0.5 %. GEOS-Chem data assimilations are used to estimate the influence of filtering and scaling to TCCON on global CO, and tend to pull concentrations away from

the prior, and closer to the truth. We conclude with suggestions for further improving the MOPITT data products.

## 1   Introduction

Carbon monoxide (CO) is an important atmospheric trace gas. It is a tracer of pollution and atmospheric transport, and plays an important role in the atmospheric hydroxyl (OH) budget. About $2800\,\mathrm{Tg\,CO\,yr^{-1}}$ are emitted globally, with about 45 % of the emissions coming from oxidation of volatile organic compounds (VOCs—predominately methane and isoprene), about 25 %

from biomass burning, 25 % from fossil fuel and domestic fuel burning, and the rest from vegetation, oceans, and geological activity (Seinfeld and Pandis, 2006). It acts as an indirect greenhouse gas (GHG) as a minor source of $CO_2$, and by affecting OH concentrations, which in turn affects the lifetime of methane. Its per mass 100-year global warming potential is 1.9 (Forster, P. et al., 2007). The ultimate fate for 90 % of CO is oxidation by OH to form carbon dioxide, and $HO_2$. CO has an average global lifetime of about 1–3 months, with a shorter lifetime in the tropics and a longer lifetime in the Southern Hemisphere

extratropics (Lelieveld et al., 2016). The moderate lifetime of CO makes it a good tracer for both emissions and transport of pollution.

The Measurements Of Pollution In The Troposphere (MOPITT) is a Canadian instrument onboard the Terra earth observing satellite, launched December 1999. Drummond et al. (2010) describe the instrument in more detail, but briefly it is a gas correlation radiometer with near-infrared (NIR) and thermal-infrared (TIR) channels. The primary MOPITT mission goal

is to quantify CO in the Earth's atmosphere. Space-based observations of CO can provide greater spatial coverage than a few surface observations. However, space-based observations that rely on reflected (e.g., NIR) sunlight can be influenced by surface properties, airglow, clouds, and are more strongly affected by aerosol scattering than solar-viewing instruments. For MOPITT the thermal-infrared (TIR) sensitivity depends on the strength of the temperature contrast between the surface and atmosphere,





which is variable across the globe. Due to the physical limitations of passive earth nadir-viewing remote sensing, satellite instruments often have lower information content per observation than ground-based instruments (e.g., Deeter et al., 2015), especially compared to the Total Carbon Column Observing Network (TCCON), which measures atmospheric absorption of the sun's radiance. Ground-based spectrometers often have higher spectral resolution and/or coverage as well as temporal

resolution at an individual location. These differences between observing systems make intercomparisons useful to check for and reduce biases.

While MOPITT data are the longest satellite record of total column CO (Deeter et al., 2017), there are several other satellite instruments that measure column CO and we mention a few of them here. SCIAMACHY (Scanning Imaging Absorption Spectrometer for Atmospheric Chartography) onboard Envisat (Environmental Satellite) launched March 2002 was first compared

with ground based observations in 2005 (Sussmann and Buchwitz, 2005), and later compared with the larger TCCON network and found to be biased about 10 % lower (Hochstaffl et al., 2018). TROPOMI (Tropospheric Monitoring Instrument) onboard Sentinel-5 Precursor was launched October 2017 and was found to be biased 6 ppb higher than TCCON with the difference depending on location (Borsdorff et al., 2018). GOSAT-2 (Greenhouse gases observing satellite-2) was recently launched in October 2018 and TCCON will be used for its validation.

Most intercomparisons with MOPITT have used aircraft data (e.g, Deeter et al., 2014, 2017, 2019). The first systematic validation of MOPITT CO with ground-based column measurements was by Buchholz et al. (2017), who used the Network for the Detection of Atmospheric Composition Change (NDACC) mid-IR retrievals. There have been some studies to compare observations from MOPITT with data from a few (3–6) TCCON sites (e.g, Mu et al., 2011; Té et al., 2016), but this is the first to use observations from all the sites in an intercomparison with MOPITT. Continual comparisons of MOPITT observations

with other systems ensures data quality, and can be used to determine areas of improvement. This intercomparison exercise uses the MOPITT version 7 joint (V7J) product, and ground-based NIR observations from the TCCON.

The rest of this paper is summarized as follows: Section 2 describes the different instruments, systems, and datasets used in this study. Section 3 describes our effort to derive filters for MOPITT data, and to improve the single sounding accuracy and precision using bias corrections. Section 4 describes the MOPITT and TCCON comparisons, including sensitivity tests and a

comparison of averaging kernels and information content. Section 5 describes data assimilation tests where the GEOS-Chem model is used to estimate how filtering and bias correcting MOPITT data affects global fluxes. Finally we conclude in Section 6 with a summary of practical considerations in this study, along with suggestions on how MOPITT retrievals might be improved in future iterations.

## 2 Datasets

### 30 2.1 MOPITT

The MOPITT instrument onboard the Terra satellite launched December 1999 has been described elsewhere (Drummond et al., 2010). Briefly, it is a gas filter correlation radiometer with eight optical channels of which three have been used since August 2001 for CO observations, two in the TIR band (channels #5 and #7, 4.617 $\pm$0.055 $\mu$m) and one in the NIR band (channel





#6, 2.334±0.011 μm). Each channel produces an "average" (A) and "difference" (D) radiance measurement. A linear detector array in each channel allows MOPITT to make observations at four different sounding locations simultaneously. The ground field of view is approximately 22 km × 22 km for each sounding. Retrievals from among these four "footprints" or pixels were previously shown to have a bias compared to ground-based column measurements from the NDACC-IRWG (Buchholz et al.,

2017). A moving mirror scans cross track for 29 "stares" in each direction for a swath that is approximately 650 km wide, and one back and forth sweep takes approximately 26 s.

Terra is in a daytime descending (nighttime ascending) sun-synchronous orbit at an altitude of about 700 km with a local equator crossing time around 10:30 (22:30 nighttime) and an inclination angle of 98.4°. Terra makes 14–15 orbits daily with an exact repeat time of 16 days. However, with its wide swath-width MOPITT is able to achieve near global coverage every

3–4 days. MOPITT data are the longest satellite record of atmospheric CO (Deeter et al., 2017). The redundancies built in to the MOPITT mission allowed for continued measurements after a cooler failure in May 2001 eliminated one of the two optical boards and the usefulness of channels 1–4, leaving channels 5–8 (Drummond et al., 2010). The impact of other early anomalies is minor. No abrupt changes since 2001 are expected to impact the retrievals, with the possible exception of annual hot calibrations the latest of which was in March 2019, and a separate temporary cooler malfunction in July 2009. Due to the

different instrument configuration from the early record, we only include MOPITT data from 2002–2017 (inclusive) in this study.

There are different retrieval products corresponding to TIR-only (T) retrievals, NIR-only (N) retrievals, and TIR-NIR (J, joint) retrievals. We chose to make comparisons with the level 2 version 7 joint (L2, V7J) product because it should theoretically contain the most information. Deeter et al. (2014) noted that the V6 TIR-NIR product has the greatest vertical resolution, but

has large retrieval errors and bias drift. The TIR-only product has the highest stability, and the NIR-only is best at CO total column retrievals. The MOPITT retrievals are performed on a logarithmic scale due to the large variability of CO in the atmosphere (∼an order of magnitude). The state vector includes up to 10 vertical layers of $\log_{10}(VMR_{CO})$ (dry volume mixing ratios), surface temperature, and surface emissivity. Retrievals are performed on a grid of 100 hPa spaced layers up to 100 hPa (e.g., surface–900 hPa, 900–800 hPa, etc. Deeter, 2017). The top layer retrieved is 100–50 hPa, and above that the

prior is used due to low sensitivity. Above 50 hPa represents 1.2±0.4% (1 σ) of the a priori CO column (1.3 % in SH, 1.0 % in NH). Fractions of CO in this layer compared to the total column are shown in the Supplemental Material (SM) Fig. S1. The a priori is from climatological output from the Community Atmosphere Model with Chemistry (CAM-chem, Lamarque et al., 2012) and is described by Deeter et al. (2014). The a priori covariance matrix is described by Deeter et al. (2010). A total column is obtained by a weighted average of the layers, and this can be converted to a column-average dry-air mole

fraction (denoted $X_{CO}$) by dividing by the model total column of dry air included in the MOPITT V7 product that takes into account surface pressure and water content. We focus on only daytime soundings, which are defined as those with a solar zenith angle less than 80° in the retrieval. In the V7J data product the 100–0 hPa layer is an average of the 100–50 hPa and 50–0 hPa layers, and we use the 100–0 hPa values for our 100–50 hPa layer but use values that are 48 % as much for 50–0 hPa based on recommendations of the MOPITT V5 users guide.





There are a number of previous studies that have compared MOPITT with different observing systems. Because the algorithm has been improved several times since the start of the mission, here we only list validation studies on versions 6 (released in 2013), and 7 (released in 2016, used in this study). Recently version 8 was released (December 2018). Versions prior to 6 are no longer available (https://www2.acom.ucar.edu/mopitt/products, last access: 2 August 2018). Deeter et al. (2014) noticed a

bias between the MOPITT V6J column retrievals and aircraft observations of $+4.3$ ppb (assuming an average total air column density of $2.1 \times 10^{25}$ molec cm$^{-2}$, roughly 5 % for a global 80 ppb average). They noted a correlation of $r = 0.89$ between the systems, and a drift of only $0.15 \pm 0.1$ ppb yr$^{-1}$ ($\sim 0.18 \pm 0.12$ %). The V6J retrievals had an overall positive bias at the surface and 800 hPa layers, a negative bias at the 600 and 400 hPa layers, and a positive bias again at the 200 hPa layer. The bias, drift, and correlation all depended on what data products were compared. Later, the V6J profiles were compared with aircraft

measurements over the Amazon basin (Deeter et al., 2016). Limited maximum aircraft altitudes precluded column retrieval comparisons, but Deeter et al. (2016) noted maximum biases at the 800 hPa of $-27$ %.

Three studies compared ground-based remote sensing observations with those from MOPITT (Rakitin et al., 2015; Té et al., 2016; Buchholz et al., 2017). Rakitin et al. (2015) made comparisons between MOPITT V6J L3 and various ground-based remote sensing sites in Eurasia. There is significant variability in the unadjusted comparisons for different sites in their study,

which could be from the influence of averaging kernels (Rodgers and Connor, 2003), but in general MOPITT observations were larger than ground-based observations. Té et al. (2016) compared MOPITT V6J and IASI (Infrared Atmospheric Sounding Interferometer) satellite observations with ground-based observations in a urban site (Paris), a high altitude site (Jungfraujoch), and a Southern Hemisphere site (Wollongong). They noted good agreement between space and ground-based observations with slopes of 0.91–0.99, satellite observations being slightly lower. Recently, Buchholz et al. (2017) compared MOPITT V6

observations with those from 14 different ground-based NDACC sites between 78°S and 80°N, and used data from August 2001–February 2012 for comparisons with V6T, V6N, and V6J. We focus on their V6J comparison results. They found MO-PITT is generally biased high relative to the NDACC, and 11 sites have a bias less than 10 % over land. The all-station mean bias is 5.1 %, and the average correlation is $\bar{r} = 0.78$. They noted the surface type (land or water) had little effect on validation statistics. However, they did note that validation results differed among pixels, and pixel 1 has the lowest correlation while

pixel 3 has the highest.

Deeter et al. (2017) is the only systematic global validation study of the MOPITT V7 algorithm. They use aircraft measurements from the HIAPER Pole-to-Pole Observations (HIPPO) campaign, and National Oceanic and Atmospheric Administration (NOAA) aircraft flask samples primarily over North America for their validation dataset. They describe the improvements included to create the V7 algorithm. They find the V7J column observations have a smaller bias and larger $r$ (1.4 ppb and 0.93

respectively) than the V6J product (3.8 ppb and 0.89).

While L1 includes radiance bias corrections, there are no empirical bias corrections to the physics-based retrieval in the L2, V7 MOPITT products. There are retrieval anomaly diagnostics included in the L2 product, but users need to define filters to use for their particular application. For L3, V7J daytime observations where both the signal to noise ratio (SNR) of channel #5A < 1000 and channel #6A SNR < 400 are excluded (Deeter, 2017). All observations from Pixel 3 are also excluded due to



excessive and unstable noise from NIR measurements from that pixel (Deeter et al., 2015). In this study suggested filters are developed along with a bias correction.

## 2.2 TCCON

The TCCON is a global network of independently operated solar-viewing Fourier transform spectrometers (SV-FTS) operated
under a common set of standards. From measurements taken by these spectrometers, retrieved estimates of $X_{CO}$ are made (Wunch et al., 2011a, 2015). Because profiles are not a part of the TCCON data product, we focus on validating the MOPITT total columns rather than profiles. Data are quality screened by both individual site operators as well as a centralized team. From sensitivity tests perturbing the algorithm to each known source of uncertainty (e.g., priors, surface pressure, etc.), GGG2014 $X_{CO}$ errors for TCCON are below 4 % (Wunch et al., 2015).

One of the primary uses of the TCCON data has been satellite validation (e.g., Inoue et al., 2016; Kulawik et al., 2016; Wunch et al., 2017b). There are several reasons why TCCON data are considered more accurate than satellite observations, and hence a good validation source. 1) Observations are directly pointed at the sun, which increases the signal-to-noise ratio (SNR), eliminates effects of surface properties, and eliminates the effects of both airglow and aerosol scattering (e.g., Zhang et al., 2015). 2) Instruments are operated at a resolution of at least $0.02\,cm^{-1}$, which provides more information for spectral
fitting than most satellite measurements. 3) The network has been around since 2004 with contributions from many different institutions. This international collaboration has lead to many discoveries on how to reduce errors in $X_{gas}$ retrievals (e.g., Kiel et al., 2016).

Despite these advantages, the data are still subject to errors. For example, to tie to the World Meteorological Organization (WMO) in situ scale there is a 7 % scaling factor in GGG2014 for $X_{CO}$ (Wunch et al., 2015). This factor is considered large
compared to the current uncertainty in spectroscopy and there is an ongoing effort to determine if this factor is appropriate. In this study we use both the official TCCON $X_{CO}$ product, as well as a derived product without the empirical scaling factor applied. For a discussion and current comparison of unscaled TCCON data to the WMO scale, see SM Sect. S2.

We compare MOPITT with TCCON from mid-2004 through 2017. Prior to 2007 there were only four TCCON sites (Table 1). During 2007 and 2008 the TCCON grew to nine sites. Table 1 also lists the site locations and number of coincidence days
after MOPITT data are filtered.

## 2.3 AirCore

AirCore measurements are a novel way to vertically sample the atmosphere to obtain profiles of various gases and have been described elsewhere (Karion et al., 2010; Membrive et al., 2017). Briefly, a coiled tube on the order of 100–300 m long, with an inner diameter on the order of 2–5 mm is taken to altitude. One end of the tube is sealed, so during ascent it is evacuated
and on descent the tube slowly fills with ambient air. Because diffusion is slow over the long length of the tube, but fast across the 2–5 mm diameter of the tube, air from different altitudes does not mix significantly. So upon landing the vertical profile of the gas is preserved along the length of the tube, high altitudes near the closed end and low altitudes near the open end. On

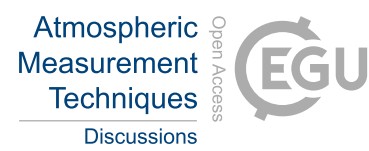

**Table 1.** Details for TCCON sites used in this study. Occasionally one site had more than one instrument as indicated by multiple 2-letter IDs.

| Site name | ID | Lat | Long | m asl | Operational dates[a] | days[b] | References |
|---|---|---|---|---|---|---|---|
| AFRC, Edwards, CA, USA | df | 34.958°N | 117.882°W | 699 | Jul 2013–present | 253 | Iraci et al. (2016a) |
| Anmyeondo, South Korea | an | 36.624°N | 126.331°E | 30 | Feb 2015–present | 24 | Goo et al. (2014) |
| Ascension Island | ae | 7.917°S | 14.332°W | 10 | May 2012–present | 151 | Feist et al. (2014) |
| Białystok, Poland | bi | 53.23°N | 23.025°E | 180 | Mar 2009–Sep 2018 | 159 | Deutscher et al. (2014) |
| Bremen, Germany | br | 53.104°N | 8.850°E | 27 | Jan 2007–present | 83 | Notholt et al. (2014) |
| Burgos, Philippines | bu | 18.553°N | 120.650°E | 35 | Mar 2017–present | 31 | Morino et al. (2018); Velazco et al. (2017) |
| Caltech, Pasadena, CA, USA | ci | 34.136°N | 118.127°W | 237 | Sep 2012–present | 56 | Wennberg et al. (2014b) |
| Darwin, Australia | db | 12.456°S | 130.927°E | 37 | Aug 2005–present | 599 | Griffith et al. (2014) |
| East Trout Lake, SK, Canada | et | 54.354°N | 104.987°W | 502 | Oct 2016–present | 28 | Wunch et al. (2017a) |
| Eureka, Nunavat, Canada | eu | 80.05°N | 86.42°W | 610 | Mar 2010–present | 8 | Strong et al. (2017) |
| Four Corners, NM, USA | fc | 36.797°N | 108.480°E | 1643 | Mar 2013–Oct 2013 | 22 | Dubey et al. (2014b); Lindenmaier et al. (2014) |
| Garmisch, Germany | gm | 47.476°N | 11.063°E | 743 | Jul 2007–present | 153 | Sussmann and Rettinger (2014) |
| Hefei, China | hf | 31.90°N | 117.17°E | 29 | Sep 2015–Dec 2016 | 17 | Liu et al. (2018) |
| Indianapolis, IN, USA | if | 39.861°N | 86.004°W | 270 | Aug 2012–Dec 2012 | 18 | Iraci et al. (2016b) |
| Izaña, Tenerife, Spain | iz | 28.3°N | 16.5°W | 2370 | May 2007–present | 65 | Blumenstock et al. (2014) |
| JPL, Pasadena, CA, USA | jc | 34.202°N | 118.175°W | 390 | Jul 2007–Jun 2008, | 10 | Wennberg et al. (2014c, 2016a) |
|  | jf |  |  |  | May 2011–May 2018 | 19 |  |
| Karlsruhe, Germany | ka | 49.100°N | 8.439°E | 116 | Apr 2010–present | 136 | Hase et al. (2014) |
| Lamont, OK, USA | oc | 36.604°N | 97.486°W | 320 | Jul 2008–present | 593 | Wennberg et al. (2016b) |
| Lauder, New Zealand | lh, ll | 45.038°S | 169.684°E | 370 | Jun 2004–present | 150 | Sherlock et al. (2014a, b); Pollard et al. (2017) |
| Manaus, Brazil | ma | 3.213°S | 60.598°W | 50 | Oct 2014–Jun 2015 | 3 | Dubey et al. (2014a) |
| Ny-Ålesund, Spitsbergen, NOR | sp | 78.923°N | 11.923°E | 20 | Mar 2006–present | 79 | Notholt et al. (2017) |
| Orléans, France | or | 47.97°N | 2.113°E | 130 | Aug 2009–present | 138 | Warneke et al. (2014) |
| Paris, France | pr | 48.846°N | 2.356°E | 60 | Sep 2014–present | 40 | Té et al. (2014) |
| Park Falls, WI, USA | pa | 45.945°N | 90.273°W | 473 | Jun 2004–present | 435 | Wennberg et al. (2014a) |
| Réunion Island | ra | 20.091°S | 55.485°E | 87 | Sep 2011–present | 275 | De Mazière et al. (2014) |
| Rikubetsu, Japan | rj | 43.457°N | 143.766°E | 380 | Nov 2013–present | 12 | Morino et al. (2014b) |
| Saga, Japan | js | 33.241°N | 130.288°E | 7 | Jul 2011–present | 136 | Kawakami et al. (2014) |
| Sodankylä, Finland | so | 67.367°N | 26.631°E | 188 | May 2009–present | 241 | Kivi et al. (2017); Kivi and Heikkinen (2016) |
| Tsukuba, Japan | tk | 36.051°N | 140.122°E | 31 | Aug 2011–present | 66 | Morino et al. (2014a) |
| Wollongong, Australia | wg | 34.406°S | 150.879°E | 30 | Jun 2008–present | 342 | Griffith et al. (2014) |
| Zugspitze, Germany | zs | 47.42°N | 10.98°E | 2960 | Apr 2015–present | 42 | Sussmann and Rettinger (2018) |

[a]Operational dates refers to time range where public GGG2014 retrievals are available. [b]Coincidence days only, and after filtering MOPITT data.





the ground, the AirCore is analyzed within a few hours, which minimizes molecular diffusion. By pulling the air through and measuring concentrations with a calibrated trace-gas analyzer, such as a Picarro, a vertical profile can be obtained.

Often AirCores are flown on balloons that can reach a ceiling of around 30 km ($\sim$10 hPa), depending on the type of balloon. Once altitude is reached, the payload is cut away from the balloon. Higher altitude data (during rapid descent) often need to 5 be discarded, hence 22 km ($\sim$40 hPa) is the median highest altitude in this dataset. The vertical resolution depends on tube design, measurement altitude, recovery time, and temperature but is on order of the 200–1000 m. From 2012–2017 there are 36 AirCore profiles available. AirCore profiles are used among other profile measurements to tie TCCON retrievals to the WMO scale (Wunch et al., 2015). Here we use them for sensitivity tests when an approximation of the true atmospheric profile is needed.

## 10  3   Quality Control Filters and Bias Correction

Typically a retrieved state vector $\hat{\mathbf{x}}$ (e.g., an atmospheric profile) is described as a linearization about the a priori state vector $\mathbf{x}_a$ (Rodgers, 2000) i.e.,

$$\hat{\mathbf{x}} = \mathbf{x}_a + \mathbf{A}\left(\mathbf{x} - \mathbf{x}_a\right) + \boldsymbol{\epsilon}_x\left(\mathbf{b}, \mathbf{c}\right). \tag{1}$$

In this equation $\mathbf{A}$ is the averaging kernel, a matrix in this case, with elements $A_{ij} = \frac{\partial \hat{x}_i}{\partial x_j}$, and $\mathbf{x}$ is the true state vector. The 15 term $\boldsymbol{\epsilon}_x$ is a catch-all for any remaining systematic or random uncertainties from instrument calibration or the retrieval. This term is a function of forward model parameters not perfectly known ($\mathbf{b}$), such as pressure, temperature, pointing, spectroscopy, and modeling of instrument response (e.g., the instrument line shape). $\mathbf{c}$ contains other values in the retrieval not used in the forward model, such as convergence criteria. Changes in $\hat{\mathbf{x}}$ may thus be related to changes in $\mathbf{b}$ and $\mathbf{c}$. To a first order, the biases in $\mathbf{b}$ and $\mathbf{c}$ have a linear effect on $\hat{\mathbf{x}}$ (Rodgers, 1990). However, these effects may not be accounted for in models so 20 measurement teams may reduce effects of these spurious variations by filtering data empirically.

For example, empirical corrections are employed for various gases in the final TCCON products after the physics based retrievals to improve accuracy up to about 0.1 % which would otherwise be currently limited to accuracies of about of 2–3 % due to spectroscopic uncertainties, especially in $O_2$ (Wunch et al., 2011b, 2015). As a second example empirical corrections to $CO_2$ measurements from the Orbiting Carbon Observatory-2 satellite (OCO-2, launched 2014) did not always improve data 25 at all scales, but did reveal areas where the algorithm could be improved (O'Dell et al., 2018; Kiel et al., 2019). Though their studies were for $CO_2$, we apply many of the same methods for CO including similar truth metrics.

By comparing retrieved data with a truth metric, some data may stand out as possibly biased due to the $\boldsymbol{\epsilon}_x\left(\mathbf{b}, \mathbf{c}\right)$ term. These may be filtered out, deweighted, or bias corrected to improve the final product. It is challenging to define a truth metric because if the true state of the atmosphere was known a priori the measurement would not be needed in the first place. Rather than 30 using metrics that work for each measurement, statistically reasonable products can be used to empirically identify artifacts and outliers. We use TCCON and a small region approximation (SRA - also called small area approximation or variation) as truth proxies. For the SRA we assume that over a sufficiently small region (e.g., $\sim$1°) remote from point sources the atmosphere is approximately homogeneous and outliers are due to inadequacies in the retrieval.



Filter selection and biases are interdependent, thus our quality control (QC) and bias correction process was iterative.

## 3.1 Pixel to pixel bias

Buchholz et al. (2017) observed biases among the four MOPITT pixels. This bias significantly affects our SRA (Section 3.2) as a biased value may be chosen as the median. We spatially grid the data in $2° \times 2°$ bins and average for each pixel separately over

monthly time-scales to evaluate variability in the bias. Here and throughout, data are averaged as described in Appendix A. We analyze multiple months, but for here show results from April and November 2016 in Fig. 1 for the difference between pixels 2 and 4. We choose these two pixels because the instrumental noise is larger for pixel 3 (Deeter et al., 2015) and pixel 1 has a known large global bias (Buchholz et al., 2017) and would therefore expect the $2-4$ difference to be a lower bound on pixel to pixel bias. We see large pixel to pixel bias polewards of $60°$. Comparing with scenes flagged as snowy or icy by retrievals from

MODIS (MOderate Resolution Imaging Spectroradiometer, also onboard Terra), we see there is some correlation between the bias with the snow/ice scenes. This bias can be positive or negative. For example, we see pixel 2 is lower than pixel 4 towards the north pole and is biased positive over Antarctica land. Over sea ice around Antarctica, pixel 2 is lower than pixel 4. We also compare pixel 1 and 3 to the weighted mean and find pixel 3 is biased low over land snow/ice and pixel 1 is biased high over both land and water snow/ice. These biases likely arise from the effects snow/ice have on the thermal contrast of the surface

and hence affect the TIR channels. For the rest of our analysis, we filter for daytime scenes and remove soundings where the MODIS diagnostics indicate the presence of any snow or ice.

     We examine temporal trends in MOPITT pixel bias compared to the weighted mean from all sensors (Fig. 2). Data are averaged globally for each pixel and surface type separately into 15-day bins. This analysis relies on the assumption that on average each pixel samples the same area. We see that the absolute bias of pixel 1 is largest. However in contrast with Buchholz

et al. (2017), we observe a negative rather than positive bias between pixel 1 and the mean in the TIR-NIR retrievals, which may be because their study was of V6 data. Pixel 3 has a smaller absolute bias that is positive. In 2002, the spread of the biases is larger than in 2017. On average, the land and water biases are similar (within 0.4 ppb), however there is a larger seasonal cycle ($\sim$1.5 ppb) for the land that may be an artifact of the sampling and averaging global 15-day bins.

     One consideration for bias corrections is whether accounting for differences in averaging kernels can account for the bias.

Buchholz et al. (2017) noticed a large absolute bias for pixel 1 as compared with NDACC observations even after accounting for averaging kernels. To examine the effects of averaging kernels we find MOPITT soundings within an ellipse ($\pm 1°$ latitude, $\pm 1.5°$ longitude) around the center location of AirCore flights. There are 22 flights with coincident observations and 1966 total corresponding MOPITT soundings. We apply averaging kernels to create simulated MOPITT column retrievals from AirCore profile measurements:

$$\hat{c} = c_a + \mathbf{a}_M^T \left( \log_{10}\mathbf{x} - \log_{10}\mathbf{x}_a \right), \qquad (2)$$

where $\hat{c}$ is the simulated $X_{CO}$, and $c_a$ is the a priori column $X_{CO}$. $\mathbf{x}$ is the dry VMR profile (from AirCore) and should not be confused with the state vector which is $\log_{10} \left( VMR \right)$ for MOPITT. For this study we have defined the MOPITT column



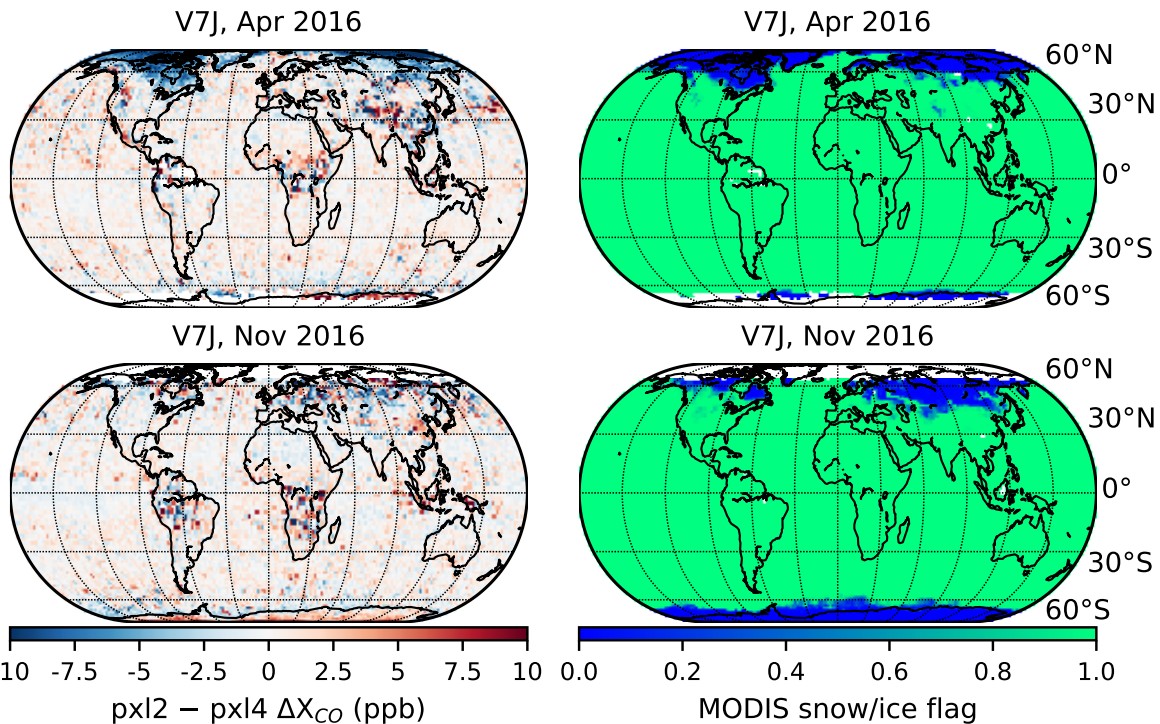

**Figure 1.** On the left are average differences in $X_{CO}$ between pixels 2 and 4. On the right is the corresponding MODIS snow/ice flag where 0 indicates all snow/ice and 1 indicates the scenes were clear of snow/ice. Some correlation is observed between bins with a large pixel to pixel bias and snow/ice cover. Here and throughout we use an Eckert IV equal area projection.

averaging kernel for a pressure level $i$ to be (Appendix B)

$$a_{M,i} = \frac{\partial \hat{c}}{\partial \log_{10} x_i} = \ln 10 \sum_{j}^{n} h_j \hat{x}_j A_{ij}. \tag{3}$$

The pressure weighting function $\mathbf{h}$ has been described by Connor et al. (2008) and Wunch et al. (2010). We find that the maximum bias is between pixels 1 and 4 and is about 5 times larger for the retrieved (2.7 ppb) than for the simulated columns (0.5 ppb, Table 2). For these soundings MOPITT is also biased high compared to the AirCore simulated columns by 3.5 ppb, which is greater than the bias of 0.5–1.4 ppb compared to other aircraft profiles (Deeter et al., 2017).

We make a preliminary pixel bias correction by adjusting soundings over land and water for each pixel separately based on a linear fit to the overall timeseries shown in Fig. 2. This fit is later improved after filtering (Sect. 3.4). After this adjustment we noticed some residual bias among the histograms, so we also apply a year to year pixel bias correction of up to 0.4 ppb that is the same for water and land.


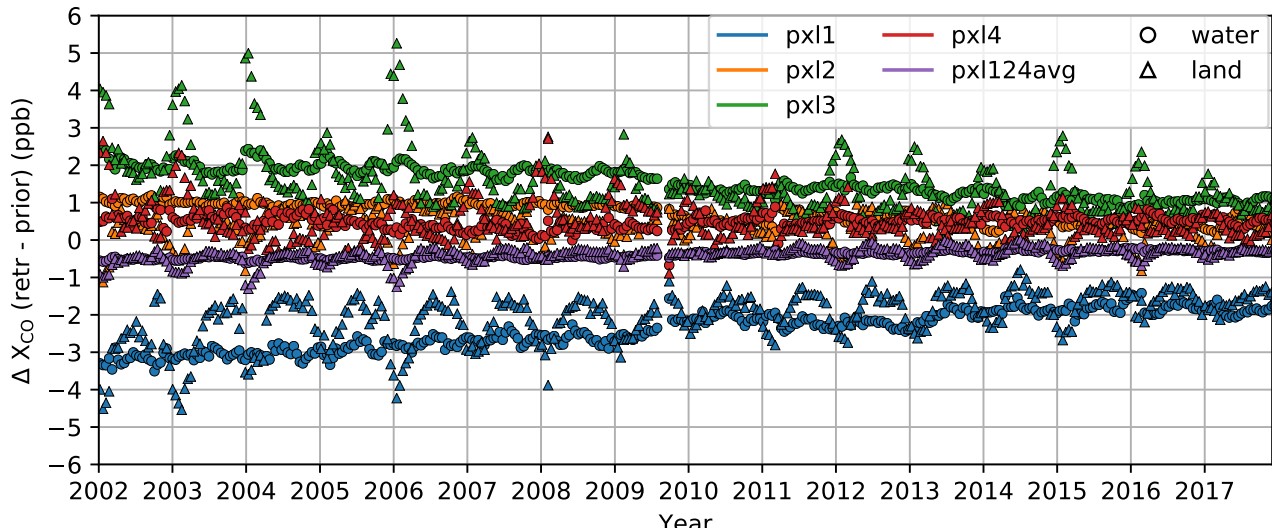

**Figure 2.** Pixel (pxl) biases compared to the weighted mean with time. Data are averaged into 15-day bins separated for land and water soundings. The mean of pixels 1, 2, and 4 is also shown because pixel 3 data are not included in the L3 product. The small gap in 2009 is from a temporary cooler malfunction on July 28.

**Table 2.** Mean values of MOPITT $X_{CO}$ retrievals colocated with AirCore measurements and separated by pixel, compared to the simulated $X_{CO}$ from applying MOPITT averaging kernels to AirCore profiles for 22 flights.

|         | Retv. mean (ppb) | Sim. mean (ppb) |
|---------|------------------|-----------------|
| Pixel 1 | 86.2             | 84.0            |
| Pixel 2 | 88.0             | 84.4            |
| Pixel 3 | 88.5             | 84.5            |
| Pixel 4 | 88.9             | 84.5            |

## 3.2 Small Region Approximation

We perform a SRA on the dataset with the preliminary filter for daytime and snow/ice free scenes and preliminary pixel bias correction. In a SRA, data within a specified area and time frame are assumed to be homogeneous, and variation within that area is assumed to be non-physical. There is always some real variation in the atmosphere, however, statistically for a large sample size these variations are expected to average out. If the area is too small then there will be too few points for an unbiased median. If the area is too large then true atmospheric variability will be significant. A disadvantage of using this method as a "truth" proxy is that it is insensitive to bias on larger scales related to e.g., latitude, and surface albedo (e.g., O'Dell et al., 2018, for OCO-2 and $CO_2$).



We use small areas that are approximately $89\,\mathrm{km} \times 133\,\mathrm{km}$ ($0.8° \times 1.2°$, latitude × longitude at the equator). Region size is a trade off between having sufficient points per region, while keeping regions small enough that real variations in $X_{CO}$ are small. The effects of different region sizes are described in the SM Sect. S3. We subtract the median from all the points within that region. If the median point does not have at least 1 degree of freedom for signal (DOF) then the entire region is discarded. We

also require at least 10 points in each region, which retains about 50 % of the SRA bins.

### 3.3 Quality Control Filters

Using the SRA "truth" proxy, we can look for correlations of differences to the local median (i.e., anomalies) with various parameters that are or may be related to the retrieval. Table 3 lists parameters we consider for filtering and bias corrections. We make plots similar to those by Wunch et al. (2011b) and O'Dell et al. (2018) (though their studies were of $X_{CO_2}$) of

anomalies versus one of the various parameters to aid in determining filter cutoffs (e.g., Fig. 3). Such plots may reveal empirical relationships with features. Similar plots with additional parameters, including some we decided were inappropriate to use as filters, are available in the SM (Sect. S4). Several features can be examined in these plots to decide on where to set the filter limits including: the underlying histograms, systematic biases from 0 in the mean including spikes, the spread among pixels - which indicates pixel-to-pixel bias, and the root mean square (RMS) from the SRA - which includes systematic and random

deviations from the truth metric. We define filters based usually on one of the following criteria 1) absolute mean bias is greater than 2 ppb, 2) RMS is greater than 6 ppb, or 3) spread of pixel-to-pixel bias is greater than 5 ppb. These criteria are not strict and we change thresholds if too few data are in a bin (due to possible sampling bias), if too many data would be removed, or if the overall trend in the mean seems like it could be corrected by a bias correction.

Several features are apparent in the SRA diagrams (Fig. 3 & SM S4–S9) that indicate data may be less reliable. For example,

there is a step change in the bias for soundings over land going from day to night. The RMS is much smaller over snow/ice free scenes (flag of 1). We also note large anomalies for low channel 5A SNR which, in agreement with the L3 product filters, suggests it is a good parameter to filter on. However, the bias is small for low channel 6A SNR soundings so unlike the L3 product we do not use it as a filter criterion. We also find the sum of the retrieval anomaly diagnostics is a better indicator for suspicious data over land than over water. These particular tests also do not support excluding all pixel 3 soundings, though on

average it does have lower and more variable DOF (Deeter et al., 2015). Maps of where data are filtered are available in the SM Fig. S10–S11. Using these filters reduces the number of daytime soundings to $3.50 \times 10^8$ (of $5.40 \times 10^8$), and reduces the RMS from 3.84 ppb to 2.55 ppb. By comparison, when we apply the L3 filters it reduces data to $3.27 \times 10^8$ daytime soundings and an RMS of 3.02 ppb.

### 3.4 Bias Correction

We observe trends in the mean bias with various parameters (e.g., Fig. 3 & SM S4–S9). To reduce the likelihood of overfitting O'Dell et al. (2018) used linear fits as bias corrections only if they removed at least 5 % of the variance for $X_{CO_2}$ from OCO-2. For $X_{CO}$ from MOPITT, the ratio between the scatter (indicated by RMS) and bias is larger than for $X_{CO_2}$ from OCO-2, however over our period of analysis there are about 400× more data over water, and about 100× more over land. Fitting



**Table 3.** Parameters in or related to MOPITT retrievals that we considered for filtering and bias correction. Data are excluded if any of the criteria below are met. Mixed surface-type soundings are also excluded.

| Field name | Source[a] | Limits land | Limits water | Description |
|---|---|---|---|---|
| $\chi^2$ | I | | | Goodness of fit indicator |
| Digital elevation model | I | | | Surface height, m a.s.l. |
| Diff. surf. emissivity (E)[b] | D | $<-0.05, >0.05$ | $<-0.75$ | $\hat{E} - E_a$, E is in state vector |
| DOF | I | $<1.05$ | $<1.00$ | $\mathrm{tr}(\mathbf{A})$ |
| Diff. surf. temperature (T)[b] | D | | $<-4$ | $\hat{T} - T_a$, T is in state vector |
| Error surface E | I | $>0.055$ | | Error on $\hat{E}$ |
| Total error $X_{CO}$ | R | | | Unused as a filter, used to weight data instead |
| Information content | D | | | $H = -\frac{1}{2}\log_2\left|\mathbf{I}_n - \mathbf{A}\right|$, unused, too similar to DOF |
| MODIS IR T threshold | I | | | MODIS cloud diagnostic 8 |
| MODIS frac. cloudy | I | | | MODIS cloud diagnostic 2 |
| MODIS snow & ice | I | $<0.999$ | $<0.999$ | MODIS cloud diagnostic 5 (see also Sec. 3.1) |
| Meas error $X_{CO}$ | R | | | Unused, see Total error $X_{CO}$ |
| Mean averaging kernel | D | $<0.75$ | $<0.50$ | $\frac{\sum \mathbf{u}^T \mathbf{A} \mathbf{u}}{n_{\mathrm{lvls}}}$, sum of all elements in $\mathbf{A}$ divided by number of profile levels ($\mathbf{u}$ is vector of ones) |
| AOD 500 nm | E | | | Average of colocated MODIS pixels (within $\pm 0.1°$) |
| Max. diff. between adj lvls | D | $>300$ | $>300$ | Indicator for possibly oscillating profiles |
| Number of iterations | I | | | Until convergence, 1–20 |
| Solar airmass | D | | | $\frac{1}{\cos(\mathrm{SZA})} + \frac{1}{\cos(\mathrm{satelliteZA})}$ |
| Smth. error $X_{CO}$ | R | | | Unused, see Total error $X_{CO}$ |
| SNR 5A (TIR) | R | $<500$ | $<900$ | L1 radiance divided by error. 1000 is a threshold for L3 TIR & TIR-NIR |
| SNR 6A (NIR) | R | | | See SNR 5A. 400 is a threshold for L3 NIR & TIR-NIR |
| SNR 7A (TIR) | R | | | See SNR 5A |
| Sum retr anom diagnostic | D | $>0.5$ | | Sum of five flags, user's guide suggests caution/exclusion if $\geq 1$ |
| Solar zenith angle (SZA) | I | $>80°$ | $>80°$ | $>80°$ is considered nighttime in the retrieval algorithm |
| tr total retr cov matrix | D | $>0.0170$ | $>0.0168$ | $\mathrm{tr}\left(\hat{S}\right)$ |
| tr meas err retr cov matrix | D | $>0.0055$ | | $\mathrm{tr}\left(\hat{S}_{merr}\right)$ |
| tr smooth err retr cov matrix | D | | $<0.0114$ | $\mathrm{tr}\left(\hat{S}_{smth}\right)$ |

[a] Source I=included in L2 files, R=ratio within L2 files, D=other derivation from L2 files, E=external. [b] Difference from the prior.
AOD = aerosol optical depth. tr = matrix trace.

concerns here are primarily over how representative the SRA is as a truth proxy, and how much the biases would already be accounted for by adjusting individual soundings using averaging kernels.

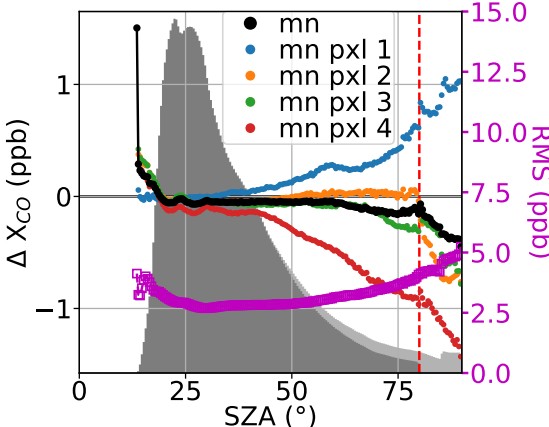

**Figure 3.** Example diagram showing the Small Region Approximation (SRA) bias as a function of solar zenith angle for water. The black points show the overall mean bias (minimum 2000 points), the magenta points show the RMS, and the other points show the mean bias for the individual pixels (minimum 300 points). The lighter histogram is of all the data. The darker histogram is data remaining after the SZA and snow/ice filters. Figures like this are used to make filters for and check for bias in MOPITT L2 data. The red line is the filter cutoff at 80°. The equivalent diagram for land, along with diagrams for other features are in the SM Sect. S4.

Even with a criterion of only 3 % reduction in the overall RMS, the only parameter to meet this is the maximum difference between adjacent levels over land. Following O'Dell et al. (2018) we make piecewise linear fits over two regimes, split at 100 ppb. The Multivariate Adaptive Regression Splines (MARS) algorithm could also be used to make a piecewise linear fit over a multidimensional dataset. However, it is more likely to overfit the data. When we applied it to the top three most variable

fields the RMS for land soundings was not significantly reduced compared with our piecewise fit so we did not use those results.

In addition to the single "feature" bias correction above, we apply a pixel-to-pixel bias correction after the filtering described in Sect. 3.3. We perform a second SRA on the filtered data without a pixel bias correction. SRA data are binned separately for each pixel and land or water surface type and averaged over 10 days. On July 28, 2009 one of the coolers on MOPITT malfunctioned, which caused a 2 month instrument shut down. We separate the period before and after this event, and make

16 different linear fits of bias relative to the all-pixel mean with time ($2\times$ land and water, $4\times$ pixel, $2\times$ time) following the method of York et al. (2004). These linear fits are used to define the pixel-to-pixel bias.

## 4    Comparisons with TCCON

### 4.1    Coincidence criteria

Various coincidence criteria have been used to match MOPITT soundings with other data sets, such as aircraft measurements,

other satellites, or ground-based sensors. For example, Deeter et al. (2014, 2017) used a co-location radius of 50 km for aircraft profiles primarily over North America, and a co-location radius of 200 km for aircraft profiles primarily over remote ocean.





Over the Amazon, Deeter et al. (2016) also used a co-location radius of 200 km, and a co-location time of 24 hrs. Té et al. (2016) used criteria of $\pm 0.15°$ latitude and $\pm 0.23°$ longitude, corresponding to 33 km×33 km over Paris. Buchholz et al. (2017) used a 1° radius and ground-based measurements within the same day. Criteria could also include fields such as the temperature of the free troposphere (e.g., around 700 hPa, Wunch et al., 2011b; Nguyen et al., 2014). Sha et al. (2018) used a sampling cone based on the solar azumith angle at the time of measurement for comparing TCCON with TROPOMI. This is likely unimportant for MOPITT given the larger footprint size (22 km×22 km versus 7 km×7 km). For example, at 60° SZA for a MOPITT pixel centered on a TCCON site at sea level, the TCCON ray would leave the MOPITT pixel at around 11 km or above 250 hPa. For a comparison of SCIAMACHY (Scanning Imaging Absorption Spectrometer for Atmospheric Chartography) with NDACC/TCCON, Hochstaffl et al. (2018) found it was necessary to deweight observations that were further away in time and space from points of comparison. This is likely much less of an issue for this study due to differences in retrieval errors and coincidence scales. For MOPITT the median retrieval error is about 3.5 ppb versus 24.8 ppb for SCIAMACHY. For SCIAMACHY temporal averaging was on order of a month compared to this study where we only use TCCON observations within $\pm 30$ min. We apply spatial averaging to the MOPITT data typically over areas of $2° \times 4°$ (with exceptions noted below). Spatial weighting is not as much of a concern here as for Hochstaffl et al. (2018) with SCIAMACHY because they used coincidence criteria of 500–2000 km radii, which are significantly larger in terms of area (about 8–100×). However, despite using smaller areas heterogeneities in CO sources that MOPITT averages over may occasionally introduce bias for real reasons (e.g., Lindenmaier et al., 2014).

We make exceptions to the $\pm 1°$ latitude $\pm 2°$ longitude spatial coincidence criteria for several sites. For sites poleward of 60° (eu, sp, and so) we expand the area to $4° \times 8°$ because the atmosphere is expected to be well mixed and retrievals are more sparse. For sites in the Los Angeles basin (ci, jc, and jf) we limit the area to 33.4–34.3°N, 116.7–118.8°W because we expect $X_{CO}$ within the basin to be much larger than the surrounding area due to urban emissions. We set the minimum latitude to 34.5°N for the AFRC site to avoid the polluted Los Angeles basin. We average soundings over land and water separately.

Because of the long (13+ year) comparison between MOPITT and TCCON, random representation error is much less important than systematic error. Té et al. (2016) and Buchholz et al. (2017) noted that systematic biases can arise from comparing total column observations (in molec cm$^{-2}$) from MOPITT and NDACC when the surface altitudes differ significantly. This effect will be diminished in column averages ($X_{CO}$) in locations away from strong local surface fluxes, however different surface altitudes can lead to biases because CO profiles are not completely uniform. Between two TCCON sites only $\sim 10$ km apart in an urban region Hedelius et al. (2017) noted an $X_{CO_2}$ difference of nearly 1 ppm. They attributed part of this to the different site altitudes. We estimate the ratio between observations at the surface pressure of TCCON versus the surface pressure of MOPITT soundings. The total column average dry mole fraction is

$$X_{CO}, ppb = \sum h_j x_j. \tag{4}$$

The vector $\mathbf{x}$ here can be either the retrieved profile or the a priori profile. We use the MOPITT profiles because they are likely more representative of the true atmosphere than TCCON a priori profiles and apply Eq. 4 to find the retrieved and prior MOPITT $X_{CO}$ at the MOPITT sounding surface pressure. We then recalculate $\mathbf{h}$ based on the daily average TCCON site





surface pressure. When TCCON altitude is lower, the MOPITT surface level is uniformly extended. For higher altitude sites, the lowest altitude MOPITT levels are either unused ($h_j = 0$) or deweighted. We then calculate $X_{CO}$ based on TCCON surface pressure. Figure 4 shows the ratios between the MOPITT retrieved $X_{CO}$ using the TCCON surface pressure compared to the MOPITT sounding surface pressure for $10° \times 10°$ areas. Larger areas are used to get a larger variety in surface pressures. We

5   see that for the high altitude Zugspitze site this scaling is particularly large (around 15 %). Over these areas the overall scaling for all sites is $0.996 \pm 0.023$ ($1\sigma$). A scaling factor less than unity is usually due to larger CO mixing ratios near the surface than the rest of the column and a lower TCCON site pressure (Hedelius et al., 2017). In this intercomparison, we account for differences in surface pressure using the $h$ vector.

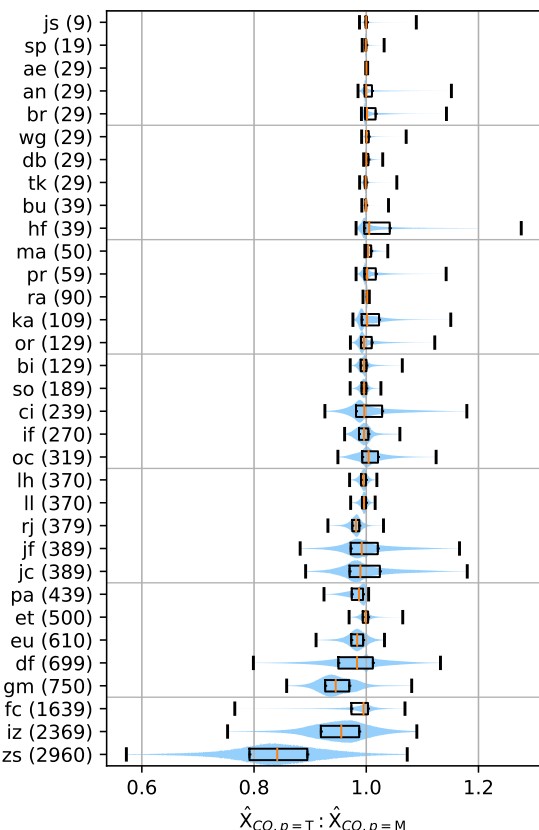

**Figure 4.** Scaling factors for MOPITT retrieved profiles if the surface were at the surface of the TCCON site (listed in m asl in parenthesis). Ordered by increasing site altitude. Soundings within $\pm 5°$ latitude and $\pm 5°$ longitude of TCCON site are used. The center 99 % of data are shown. Blue filled sections indicate data density, similar to a violin plot but using histograms rather than a kernel density estimation due to sufficient data. Black boxes indicate the central 50 % of data, and medians are orange.





## 4.2 Overall comparison using different methods

MOPITT and TCCON use different priors and have different averaging kernels (AKs, Sect. 4.4) and these differences in sensitivity need to be taken into account when comparing retrievals from the different instruments. Retrievals are also on different vertical grids, and regridding is described in Appendix C. There are different approaches to account for different priors and AKs such as the choice of comparison ensemble. Here, we choose method II detailed in Appendix D as our standard choice, and it is equivalent to how Wunch et al. (2011b) accounted for differences in a priori profiles and AKs when comparing $X_{CO_2}$ between GOSAT (Greenhouse Gases Observing Satellite) and TCCON. Figure 5 shows the comparison using this method. Supplemental Fig. S12 is a series of bar-charts of how this method compares with the others for each site. Figure 6 shows the comparison for a variety of tests when AKs are applied differently or not at all, for different spatial coincidence criteria, when filtering or bias corrections are not applied, and with a "TCCON" product without empirical corrections for airmass and to the WMO scale (Wunch et al., 2015). Generally all comparisons show MOPITT is about 6–9 % higher than TCCON, with some exceptions. This is similar to the 5.1 % positive bias between MOPITT V6J and NDACC total column observations (Buchholz et al., 2017). For method III, where AKs are applied in a manner opposite to method II, the bias is as high as 15 % but typically is closer to ∼10 % or less. In Fig. 6g where the TCCON data do not have the standard scaling to aircraft, the bias between the datasets is less than 0.5 %. Due to uncertainties in the TCCON WMO scaling (SM Sect. S2), some comparisons are made without it. Deeter et al. (2017) found a positive bias less than about 1 % when MOPITT V7J data were compared with NOAA flask measurements from aircraft.

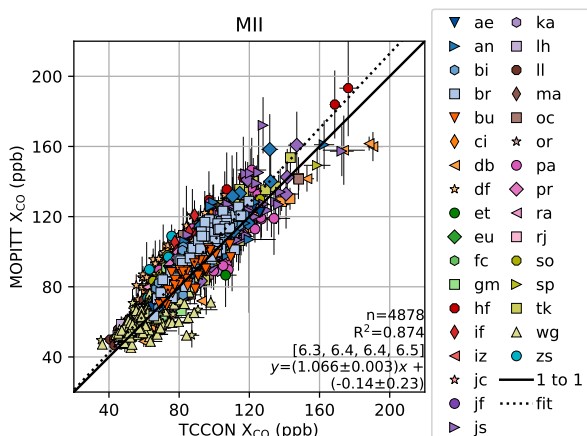

**Figure 5.** One-to-one plot comparing MOPITT and TCCON, following method II (similar to Wunch et al., 2011b, Appendix D). MOPITT data were adjusted to the TCCON a priori ($\hat{c}'_M$) and MOPITT averaging kernels were applied to TCCON data ($\hat{c}_{M \leftarrow T}$). Error bars represent standard deviations of the weighted averages. Triangles represent soundings over water, other shapes are over land. Text is number of points or days $n$, coefficient of determination for ordinary least squares regression $R^2$, bias (in %) at 50, 75, 100, and 150 ppb using the shown fit, and equation for the shown fit using the methods of York et al. (2004).



**Figure 6.** See Table D1 for a list of different methods with TCCON data on the x-axis, and MOPITT on the y-axis. See Fig. 5 for a description of the text. Unless specified, derived filters and bias corrections for MOPITT are applied. Colorbars indicate number of points in heatmap bins. **(a)** Using method III. **(b)** Using a variety of simulated profiles at "truth" to estimate TCCON and MOPITT retrieval errors (see Sect. D2). **(c)** Using method 0 (i.e., not accounting for averaging kernels). **(d)** Same as Fig. 5 but with half-sized spatial bins. **(e)** Same as Fig. 5 but with double-sized spatial bins. **(f)** Same as (c), but scaling MOPITT data to account for differences in surface pressure compared with the TCCON site (Fig. 4). **(g)** Using TCCON data without an overall scaling factor to tie to the WMO scale, and without empirical corrections for airmass. **(h)** Without bias corrections for MOPITT observations. **(i)** Without filtering MOPITT observations.





Figure 7 shows boxplots of the MOPITT to TCCON differences for each site for land only and water only soundings. We do not note an overall bias between land and water. For all sites the TCCON−MOPITT bias is positive and usually on order of about 3–10 ppb with a few exceptions. For example, MOPITT observations compared to the AFRC (df) TCCON are particularly high (∼14 ppb). This could be related to the high albedo or high surface temperatures of this desert site.

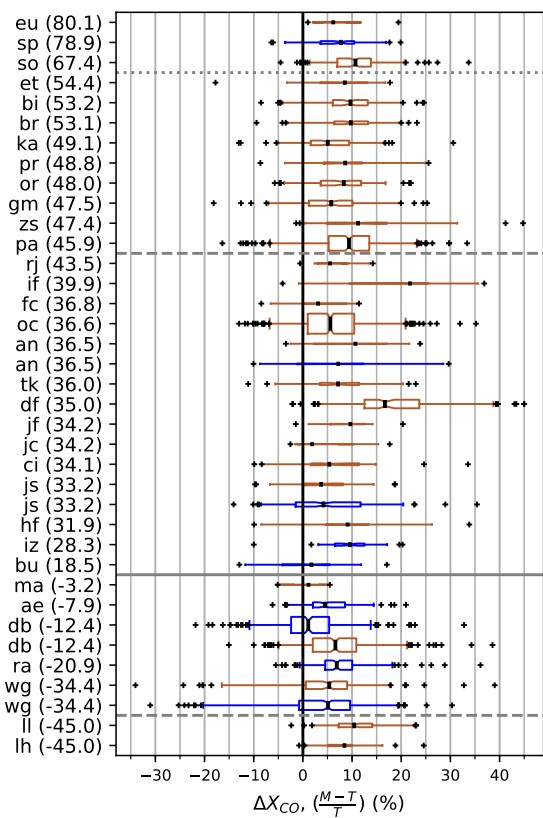

**Figure 7.** Boxplots of the MOPITT-TCCON percent difference at the TCCON sites, ordered by latitude (degrees north in parenthesis). Blue boxes are MOPITT soundings over water, brown boxes are over land. Whiskers represent the inner 95 % of data. Notches are 95 % confidence intervals of the median. Box heights represent the relative number of observations. The solid horizontal line is the equator, dashed lines are $\pm 45°$, and the dotted line is 60°N.

## 4.3 Systematic biases

A seasonal variation in bias may be indicative of differences in sensitivities between the instruments to some feature, such as airmass or water content, that varies seasonally. Figure 8 shows the timeseries of the difference averaged in 1 month 5° latitude bands. We do not find a clear long term trend in the MOPITT−TCCON difference. Deeter et al. (2017) reported a bias drift of $-0.04 \pm 0.10 \% \, \text{yr}^{-1}$ for V7J though bias drifts for individual layers were larger. Including a correction trend to the L1 radiances significantly reduced the bias drift for the layers (Deeter et al., 2019). Seasonalities of the difference for each site are





in SM Fig. S13. There does not appear to be a persistent seasonal trend for all sites, though there is some seasonal variability for individual sites. For Lamont and AFRC the bias is larger in July–October, while for Białystok the bias is larger April–June. At Ascension the bias is largest January–February, while for Reunion it is largest September–November. We do not make a seasonal bias correction.

There appears to be some latitudinal dependent bias, with a larger bias in the Northern Hemisphere. Part of this could be related to stratospheric CO (SM Sect. S1). Deeter et al. (2017) also showed some latitudinal variation in MOPITT retrievals compared with aircraft. They suggested that part of the variability could arise from interfering species such as $N_2O$, which has spectral lines that overlap the TIR channels. Before V7 a constant value of $N_2O$ was assumed, which was determined to cause biases on order of a few ppb (Deeter et al., 2017). In V7 a global average is used based on a linear fit to monthly in situ

observations. Supplemental Fig. S14 shows the bias as a function of column $N_2O$ measured by the TCCON. There is a slope of $-0.40 \frac{X_{CO\,ppb}}{X_{N_2O\,ppb}}$ though the overall correlation is small ($R^2=0.08$). There also appears to be a small dependence on column $H_2O$, which was likely reduced in V8 (Deeter et al., 2019). We do not make bias corrections for any of these systematic features.

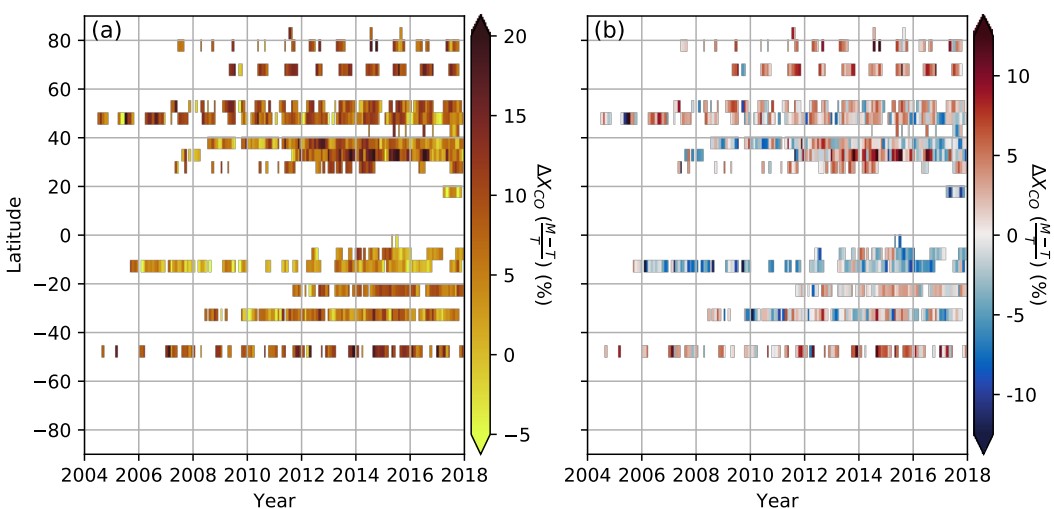

**Figure 8.** Rotated Hovmöller diagram of the mean percent differences between binned MOPITT and TCCON data using method II (Appendix D). Latitude bins are 5° zonal bands, and temporal bins are monthly. Left plot is using standard TCCON data. Right plot is without the TCCON scaling to aircraft (WMO scale - see SM Sect. S2).

## 4.4    Averaging kernels, covariance matrices, and information content

According to Rodgers and Connor (2003, Sect. 2 therein), an intercomparison of two observing systems should also include a

comparison of (1) averaging kernels, (2) retrieval noise covariance, (3) degrees of freedom, and (4) the Shannon information content. In conjunction with (1), we think it is also helpful to compare a priori profiles, which is done in Appendix E. Because





the MOPITT retrievals are of logarithmic profiles and the TCCON uses a linear scaling retrieval some aspects of the comparison are inherently different.

Example AKs for MOPITT and TCCON are shown in Fig. 9. Because the MOPITT retrieval is on a log-scale, we make an assumption that the a priori VMRs represent the true profile to obtain unitless AKs (Appendix B). We find the TCCON AKs
are more sensitive than MOPITT. Shaded regions in Fig. 9a show a wide variability in MOPITT column AKs. In addition, the typical state significantly affects the MOPITT AKs (e.g., compare Pasadena and Lauder). TCCON CO column AKs are most sensitive to the stratosphere, and are assumed to be consistent at all sites. We make a sensitivity test where the AKs were explicitly calculated in GGG2014 for days with a wide range of $X_{CO}$ at the East Trout Lake site. In general the difference from the standard AKs is small, on the order of 5 % at most.

Priors and MOPITT retrieved profiles along with their differences for select sites are shown in Appendix E. We compare MOPITT and TCCON priors. In general, MOPITT priors are influenced more by localized emissions as they are based on $1°$ simulated monthly climatologies from the CAM-chem model (Deeter et al., 2014). This can be seen especially at Pasadena, and to a lesser extent at Lamont and Tsukuba. Ascension Island shows a special case where enhanced CO in the lower free troposphere is seen coming from biomass burning and rain forest VOC emissions in Africa. At sites far removed from local
emissions (e.g., Ny-Ålesund and Lauder) the MOPITT and TCCON priors are in better agreement with each other (see e.g., Pollard et al., 2017).

Rather than compare the retrieval noise covariance, we compare reported errors and measures of precision and accuracy. Histograms of total reported retrieval error for MOPITT are shown in SM Figures S4o and S5o. With our prescribed filtering, global mean uncertainty values are $2.60 \pm 1.27 \,(1\sigma)$ ppb for smoothing, $2.68 \pm 1.40 \,(1\sigma)$ ppb for measurement, and $3.86 \pm$
$1.63 \,(1\sigma)$ ppb for the total error. The average of the errors reported in the TCCON files is $0.62 \pm 0.50 \,(1\sigma)$ ppb. However, these errors are more a measure of repeatability rather than the total error or the accuracy. The $2\sigma$ uncertainty for TCCON (GGG2009) was reported as 4 ppb (Wunch et al., 2010), and the uncertainty budget from a range of sensitivity tests is less than 4 % (Wunch et al., 2015).

Histograms of the MOPITT DOF for signal for water and land are shown in Figures S4d and S5k. The DOF for signal ($d_s$)
can be determined from

$$d_s = \mathcal{E}\left\{(\hat{\mathbf{x}} - \mathbf{x}_a)^T S_a^{-1} (\hat{\mathbf{x}} - \mathbf{x}_a)\right\}, \tag{5}$$

where $\mathcal{E}$ is the expected value operator (Rodgers, 2000, Eq 2.46 therein). However, $d_s$ is usually determined from the trace of the averaging kernel matrix (Rodgers, 2000, Eq 2.80 therein), which is equivalent to Eq. 5 for profile retrievals. Because GGG2014 is a scaling retrieval, we treat TCCON measurements as having $d_s = 1$. With a profile retrieval we would expect
$d_s > 1$, as was the case for $CO_2$ (Connor et al., 2016).

Finally, the information content $H_s$ is a measure of how accurate a measurement is to how well a value is a priori known. Rodgers (2000) expresses it on a natural log scale (Eq. 2.73 and 2.80 therein)

$$H_s = \frac{1}{2}\ln\left|\hat{\mathbf{S}}^{-1}\mathbf{S}_a\right| = -\frac{1}{2}\ln\left|\mathbf{I}_n - \mathbf{A}\right|, \tag{6}$$

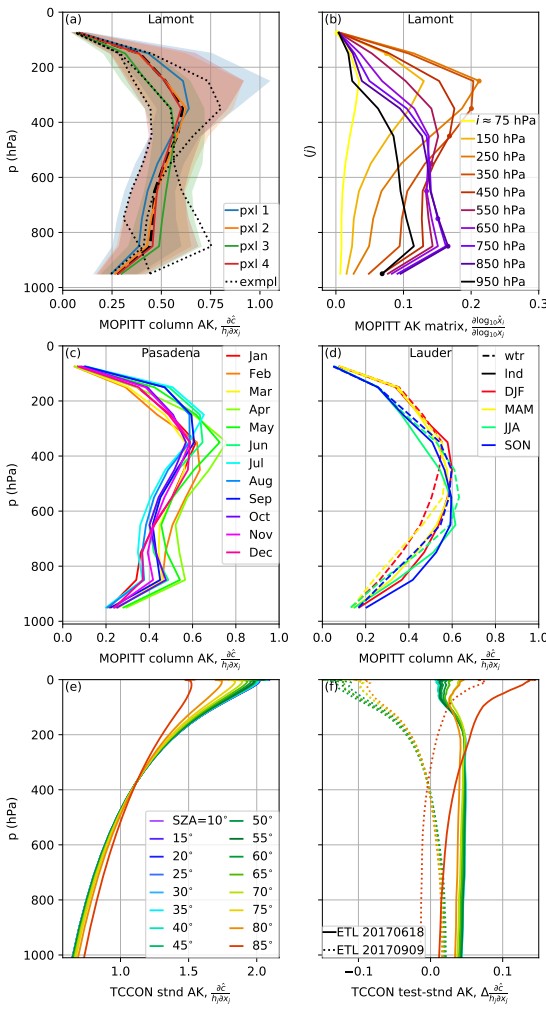

**Figure 9.** Examples of AKs from TCCON and MOPITT—subplots are not always related. MOPITT daytime AKs (a–d) are shown as center-point values along the y-axis for clarity, though MOPITT retrievals are layer averages. Unitless MOPITT column AKs are generated using the methods of Appendix B. (a) MOPITT column AKs around Lamont for 2012–2013 separated by pixel. Filled areas are the central 80 % and solid center lines are the medians per level. Black lines are select examples from single soundings that show wide variability from sounding to sounding. The thicker example corresponds to the full AK in subplot b. (b) Example MOPITT full AK from 16 Oct 2012. Dots highlight the $i$th level shown in the legend. (c) Median MOPITT column AK per level by month for Pasadena. (d) Median MOPITT column AK per level by season for land and water soundings for Lauder. (e) Standard TCCON GGG2014 AKs, which are assumed to be a function of only SZA and pressure. (f) Differences from AKs explicitly calculated at the ETL site on specific days compared to the standard AKs. For 18 Jun 2017, the mean of the ETL $X_{CO}$ is $74.0 \pm 0.8 \, (1\sigma)$ ppb. For 9 Sep 2017, the mean of the ETL $X_{CO}$ is $169 \pm 32 \, (1\sigma)$ ppb with a range of 95–225 ppb.





where $\mathbf{I}_n$ is the identity matrix. Here we express $H_s$ on a $\log_2$ scale instead. Histograms of $H_s$ for MOPITT profile retrievals are shown in Figs. S6(a) and S8(a) in the SM and values are on the order of 2.5–5.5 bits over water and 2.5–7 bits over land. If model values of $X_{CO}$ are accurate to about 32 ppb, and if the TCCON accuracy is about 4 ppb then the TCCON $X_{CO}$ information content is about $\log_2 \frac{32}{4} \approx 3$ bits.

## 5 Model assimilations

We assimilate MOPITT observations using the GEOS-Chem model to show how filtering and the bias correction affect estimated emissions inferred from inversion analyses. We conducted three experiments in which we assimilated the following datasets: 1) the original MOPITT data, 2) the filtered and bias-corrected data with a scaling to the standard TCCON data (Fig. 5) (referred to as Assim. 2), 3) the filtered and bias-corrected data with a scaling to the TCCON-based data not tied to the WMO scale and without the empirical airmass correction (Fig. 6g) (referred to as Assim. 3). The assimilation is performed using the GEOS-Chem four-dimensional variational (4D-Var) data assimilation system, employing version 35J of the adjoint model at a horizontal resolution of $4° \times 5°$. The GEOS-Chem 4D-Var system has been used in previous studies for assimilation of MOPITT data (e.g., Kopacz et al., 2010; Jiang et al., 2013, 2015, 2017). We assimilate the MOPITT data to optimize monthly average CO emissions. We assimilate daytime observations for the periods of October–December 2009, and May–July 2011 to coincide with flights from the HIAPER Pole-to-Pole Observations (HIPPO) campaign. The posterior CO fluxes are compared with the a priori, and the posterior CO concentrations are validated against CO measurements from the HIPPO 10-second merged data (Santoni et al., 2014).

The assimilation uses the offline CO simulation in GEOS-Chem with prescribed monthly mean OH fields from TransCom (Patra et al., 2011) to compute the sink of CO. The prior anthropogenic CO emissions are from the EDGAR v4.2 inventory, which are overwritten regionally with the following inventories: the Streets 2006 emissions over China and Southeast Asia from Zhang et al. (2009), the annual Canadian anthropogenic emissions from the Criteria Air Contaminants (CAC), the National Emissions 2005 Inventory (NEI2005) from the United States Environmental Protection Agency (EPA), the Co-operative Programme for Monitoring and Evaluation of the Long-range Transmission of Air Pollutants in Europe (EMEP) inventory, and the Big Bend Regional Aerosol and Visibility Observational (BRAVO) inventory in Mexico. The Global Fire Emissions Database version 3 (GFED3) provides the biomass emissions. The biofuel emissions are the Yevich and Logan (2003) inventory. The initial condition of CO states is generated by spinning up the GEOS-Chem model from January 2009. The initial CO concentrations are not optimized in the assimilation. The prior emissions are scaled by a factor of 1.5, and the emission error is purposely set to be 500%, so that the posterior CO source estimates will be less influenced by the a priori emissions and more strongly reflect the information from the filtered MOPITT observations.

Compared to the HIPPO-2 and HIPPO-4 measurements, the a priori are biased low by approximately 5% (Tables 4 and 5). In general, the original MOPITT data assimilation and assimilation 3 tend to agree with each other as compared with HIPPO. Assimilation 2 results are lower. This suggests that scaling down MOPITT observations to match TCCON is translated to less





**Table 4.** Comparisons of GEOS-Chem model simulated mole fractions assimilating MOPITT data with HIPPO-2 observations. Uncertainties are $1\sigma$. Units are in ppb. See text for descriptions of the assimilations.

| Latitude bands | Prior | Orig. MOPITT | Assim. 2 | Assim. 3 |
|---|---|---|---|---|
| $[-40, -20]$ | $-4.8 \pm 14.0$ | $2.5 \pm 15.3$ | $-2.4 \pm 14.9$ | $2.1 \pm 15.2$ |
| $[-20, 20]$ | $-5.4 \pm 6.9$ | $0.0 \pm 7.3$ | $-6.0 \pm 6.9$ | $-0.3 \pm 7.3$ |
| $[20, 40]$ | $-5.3 \pm 22.7$ | $14.0 \pm 28.3$ | $-4.1 \pm 22.9$ | $13.2 \pm 28.1$ |
| $[40, 60]$ | $-7.4 \pm 14.1$ | $2.9 \pm 15.7$ | $-6.2 \pm 13.9$ | $2.5 \pm 15.6$ |
| $[-90, 90]$ | $-5.2 \pm 19.0$ | $3.3 \pm 19.9$ | $-4.2 \pm 18.9$ | $3.0 \pm 19.8$ |

CO in the assimilation, as expected. However, the difference depends on the specific latitude band as does the assimilation that is closest to HIPPO.

To validate the quality of the filtered and bias-corrected MOPITT observations, the prior CO fluxes are compared with the a posteriori (Fig. 10). Again we find that assimilations 1 and 3 are in general agreement and assimilation 2 produces lower

5 fluxes. Fluxes using assimilated data are nearly always smaller than fluxes using the prior scaled up by 50%. Assimilation 2, which includes scaling MOPITT to the standard TCCON product, returns results that are the most similar to the unscaled prior and is higher by about 30% and 15% during HIPPO-2 and HIPPO-4 respectively.

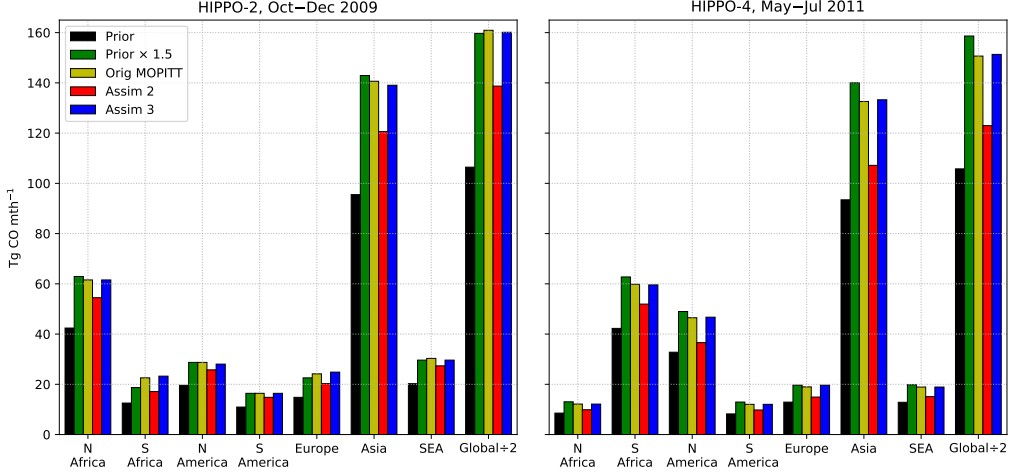

**Figure 10.** Emission estimates from assimilations in GEOS-Chem adjoint model for two different times. For this figure global emissions are scaled down by 2. SEA=South East Asia.



**Table 5.** Comparisons of GEOS-Chem model simulated mole fractions assimilating MOPITT data with HIPPO-4 observations. Uncertainties are $1\sigma$. Units are in ppb. See text for descriptions of the assimilations.

| Latitude bands | Prior | Orig. MOPITT | Assim. 2 | Assim. 3 |
|---|---|---|---|---|
| $[-40, -20]$ | $-2.0 \pm 5.4$ | $2.3 \pm 5.9$ | $-0.5 \pm 5.4$ | $2.4 \pm 5.8$ |
| $[-20, 20]$ | $-6.8 \pm 6.0$ | $-1.5 \pm 6.5$ | $-5.5 \pm 6.4$ | $-1.4 \pm 6.5$ |
| $[20, 40]$ | $-13.2 \pm 17.9$ | $-2.9 \pm 18.9$ | $-8.6 \pm 18.2$ | $-2.7 \pm 19.0$ |
| $[40, 60]$ | $-3.5 \pm 27.3$ | $12.1 \pm 26.6$ | $4.2 \pm 27.1$ | $12.4 \pm 26.6$ |
| $[-90, 90]$ | $-4.1 \pm 17.9$ | $6.0 \pm 19.2$ | $0.5 \pm 18.5$ | $6.3 \pm 19.3$ |

**Table 6.** Summary of practical considerations comparing MOPITT and TCCON soundings.

| Practical consideration | Example | This work |
|---|---|---|
| Coincidence criteria | Nguyen et al. (2014) | $2° \times 4°$, $\pm 30$ min (typically), Sect. 4.1 |
| Weighted averaging | Hochstaffl et al. (2018) (distance & time) | On reported errors $\frac{1}{\epsilon^2}$, Apdx A |
| Weighted avg error | Gatz and Smith (1995) | Gatz & Smith method, Apdx. A |
| Representation errors (e.g., $p$) | Buchholz et al. (2017); Té et al. (2016) | Accounting for surface $p$, Sect. 4.1 |
| Filtering | Genetic algorithms, e.g., Mandrake et al. (2013) | Chosen using SRA, Sect. 3.3 |
| Bias corrections | Wunch et al. (2011b); O'Dell et al. (2018) | Primarily pixel-based, Sect. 3.4 |
| Accounting for different $\mathbf{x}_a$, $\mathbf{A}$ | Rodgers and Connor (2003); Wunch et al. (2011b) | Appendix D |
| Pre-averaging | Buchholz et al. (2017), Apdx. B therein | Typically point-wise application of $\mathbf{x}_a$, $\mathbf{A}$ |
| Vertical regridding | Delhez (2003) | Mass conserving, Apdx C |

# 6   Conclusions

## 6.1   Practical considerations in intercomparisons of remote sounding retrievals

In addition to the formal aspects of intercomparing retrievals from different remote sounding retrievals, there are a variety of practical aspects to consider. For several of these aspects, an entire study could be devoted to them for each intercomparison. We summarize our comparison methodology in Table 6 and give examples of other studies that provide additional details or alternative methods. Though it is impractical to test all combinations of different considerations, we test some as described in Sect. 4.2 such as coincidence criteria, filtering, bias corrections, and applications of averaging kernels.

## 6.2   Considerations for future MOPITT data use

Several lessons learned in this study may be useful for future versions of MOPITT data products or users assimilating the data. Additional fields used in the retrieval, such as the a priori mixing ratio from 50–0 hPa and the water vapor profile would be





useful outputs when converting mixing ratios from whole-air to dry-air. Though the prior covariance matrix is fixed (Deeter et al., 2010), a single matrix per daily file may be helpful. The retrieved surface emissivity over land is on average about 0.007 or about 0.75 % larger than the prior, and the retrieved surface temperature is on average about 6 K larger than the prior (see histograms SM Fig. S5f, S5m)). This suggests prior values should perhaps be reconsidered as they may be biased low over
land. Further updates to prior values of CO, $N_2O$, and $H_2O$ are expected to further improve the retrievals. For example, the retrieved column CO is slightly larger than the a priori globally (SM Fig. S15) but the difference depends on the level and could be related to uncertainties in model transport, sinks, and sources.

     Filtering can reduce spurious values. MOPITT files include parameters that could be used in filtering such as a Retrieval Anomaly Diagnostic, various cloud indicators, and DOF. Data users should consider creating a QC flag for their analyses or a
binary flag could be included in future versions, e.g. based on parameters in Table 3 or based on the recommendations of the MOPITT team (e.g., the L3 filters). Often highly deviant retrieved surface temperatures show up around coastlines (especially western coastlines, SM Fig. S10) that did not pass our quality screening. These may be related to sounding definitions of surface type. SNR 6A is used to filter MOPITT data when creating the TIR-NIR L3 product to maintain consistency with the NIR product and increase stability of the DOF, but we do not find sufficient evidence to use it as a TIR-NIR filter criterion
based on $X_{CO}$ stability alone.

     When biases are found in the MOPITT L2 data the strategy is to correct the L1 radiances or the retrieval algorithm (e.g., Deeter et al., 2019). MOPITT data users of L2 $X_{CO}$ may consider implementing a bias correction before analysis or model assimilation. In terms of $X_{CO}$, pixel 3 data agree with pixels 2 and 4, however this agreement may not necessarily hold for retrieved profiles and pixel 3 data are excluded in the L3 product due to excessive NIR noise and to increase stability in the
DOF (e.g., Deeter et al., 2015). A bias correction should be considered when assimilating pixel 1. There is a bias in the SRA for large retrieval errors on $X_{CO}$ above about 8 ppb (SM Fig. S4o, S5o). This bias suggests that perhaps these data should be excluded or deweighted further, which we did not do here. A bias adjustment field could also be included as a field in future MOPITT files. Such an adjustment could account for empirical biases noted with various parameters, pixel-to-pixel biases (Sect. 3.4), and an overall bias compared with NDACC (Buchholz et al., 2017), aircraft flights (Deeter et al., 2017), and/or
TCCON.

*Data availability.*  MOPITT data were obtained from the NASA Langley server (ftp://l5ftl01.larc.nasa.gov/MOPITT/, last access: 12 December 2018). TCCON data were obtained through the TCCON data archive hosted by CaltechData (https://tccondata.org/, last access: 12 December 2018). See Table 1 for data references for each site. TCCON data without the scaling to the WMO scale were obtained from the site PIs. AirCore data were obtained from Colm Sweeney (v20170918).




**Appendix A: Calculation of $X_{CO}$ and weighted averaging**

The MOPITT V7 data product contains fields for retrieved total column CO (in molec cm$^{-2}$). Unlike the TCCON, MOPITT does not retrieve a dry-air column. However, a model dry air column is provided. We obtain a dry-air mole fraction from

$$X_{CO}, ppb = \frac{retrieved\ CO\ column}{model\ dry\ air\ column} \times 10^9. \tag{A1}$$

5 The retrieval error in molec cm$^{-2}$ can be converted to ppb the same way.

When averaging $n$ soundings together, we use a weighted average using the inverse squared retrieval errors as weights. The average retrieved value $\bar{\hat{y}}$ is

$$\bar{\hat{y}} = \frac{\sum_i^n \hat{y}_i \hat{y}_{i,err}^{-2}}{\sum_i^n \hat{y}_{i,err}^{-2}}, \tag{A2}$$

where $\hat{y}_i$ denotes an individual measurement in the bin, and $\hat{y}_{i,err}$ is the corresponding error. When an average weighted error is needed, we calculate a weighted standard error of the mean (SEM) using

$$SEM = \sqrt{\frac{n}{(n-1)\left(\sum_i^n \hat{y}_{i,err}^{-2}\right)^2} \sum_i^n \left(\hat{y}_{i,err}^{-2}\right)^2 \left(\hat{y}_i - \bar{\hat{y}}\right)^2}. \tag{A3}$$

In the case of uniform weights $\hat{y}_{i,err}^{-2}$ this reduces to the typical SEM equation. We also test a bootstrap analysis (Efron and Gong, 1983) on binned data for one of the parameters (DEsfc) in the bias correction analysis (Section 3.4) to evaluate Eq. A3. Data are placed into 146 bins with at least 2000 points in each. The bootstrap is ran 500 times per bin. We find, in agreement 15 with Gatz and Smith (1995), that Eq. A3 is a reasonable approximation to the SEM determined from the bootstrap method with an offset of only +0.2 % ± 3.1 % (1 $\sigma$).

**Appendix B: MOPITT column averaging kernel**

We derive our own MOPITT column averaging kernel (AK) vector based on the full averaging kernel matrix. To fulfill Eq. 2 (and using Eq. 4), MOPITT AK elements $a_j$ are:

20 $$a_j = \frac{\partial \hat{c}}{\partial \log_{10} x_j} = \frac{\partial \sum_i h_i \hat{x}_i}{\partial \log_{10} x_j}. \tag{B1}$$

Making use of $\frac{\partial \log_{10} \hat{x}_i}{\partial \hat{x}_i} = \frac{1}{\hat{x}_i \ln 10}$, and $A_{ij} = \frac{\partial \log_{10} \hat{x}_i}{\partial \log_{10} x_j}$

$$a_j = \sum_i h_i \frac{\partial \hat{x}_i}{\partial \log_{10} x_j} = \ln 10 \sum_i h_i \hat{x}_i \frac{\partial \log_{10} \hat{x}_i}{\partial \log_{10} x_j} = \ln 10 \sum_i h_i \hat{x}_i A_{ij}. \tag{B2}$$

This MOPITT column averaging kernel is not directly comparable with the TCCON column averaging kernels because of the log scale. A unitless column averaging kernel can be made, but requires an a apriori assumption about the true state of the



atmosphere. E.g.:

$$\frac{1}{h_j}\frac{\partial \hat{c}}{\partial x_j} = \frac{1}{h_j}\frac{\partial \hat{c}}{\partial \log_{10} x_j}\frac{\partial \log_{10} x_j}{\partial x_j} = \frac{1}{h_j}\frac{a_j}{x_j \ln 10}. \tag{B3}$$

## Appendix C: Vertical regridding

We find it necessary to express values from one retrieval on the vertical pressure grid of the other. MOPITT profiles are reported
as layer averages, but TCCON profiles are reported as level values. TCCON profiles are converted to the MOPITT grid by
linear interpolation. We divide each MOPITT layer into 500 finer equal pressure layers (about 0.4 hPa each). We interpolate
the TCCON profiles to these finer layers and then take the overall average to put the TCCON profile on the MOPITT pressure
grid.

Basic interpolation should not be used to convert the MOPITT layer averages to the TCCON grid because it does not require
that mass is conserved when the layers have different widths. Instead we use a mass conserving linear interpolation scheme
based on the MOPITT layer midpoints. This is based in part on the work of Delhez (2003).

## Appendix D: Accounting for different averaging kernels and priors when comparing MOPITT and TCCON

### D1   Adjust to a common prior and apply AKs from one system to other

Measurements from different remote sounders (e.g., MOPITT and TCCON) are not directly comparable due to their different
averaging kernels and priors. There are various methods to account for these differences, for example the commonly used
methods of Rodgers and Connor (2003), and further detailed by Wunch et al. (2011b). Briefly, these methods are based on
choosing a comparison ensemble profile ($\mathbf{x}_c$), which is often the prior of one of the sounders. Then, one of the retrieved
profiles (which we call system 1) is taken and the averaging kernels of the other (system 2) are applied in the manner of Eq. 1.
These adjusted results are then compared with the retrievals from system 2.

This method raises the questions, which system priors should we use, and which system averaging kernels should be applied?
Arguments for using TCCON are that the profile is finer at all levels except the surface, the algorithm has less of a tendency
to overfit the radiances, and it has often been used in previous studies. An argument for using MOPITT is that the retrieval is
a profile retrieval rather than column scaling. For the choice of $\mathbf{x}_c$, the MOPITT prior is 4D, while the TCCON is only 3D.
However, if TCCON averaging kernels are applied then there is an argument for using it as $\mathbf{x}_c$. For MOPITT and TCCON
these choices are formally listed in Table D1.

### D2   Use an ensemble of "true" profiles

A difficulty to the above method is that multiple adjustments are required. For example, Method II in Table D1 involves
adjusting both the MOPITT product and the TCCON product to be as if it were MOPITT, but using the TCCON a priori. The
method also requires choosing which system to apply AKs to, which can have an effect on the comparison. Depending on $\mathbf{A}$,





**Table D1.** Equations to account for different priors and averaging kernels when comparing MOPITT and TCCON soundings.

| Method | $\mathbf{x}_c$ | x (assumed) | Comparison 1 | Comparison 2 |
|---|---|---|---|---|
| 0 | | | $\hat{c}_T$ | $\hat{c}_M$ |
| I | $\mathbf{x}_{T,\mathrm{a}}$ | $\hat{\mathbf{x}}'_M = \hat{\mathbf{x}}_M + \mathbf{x}_{T,\mathrm{a}} - \mathbf{x}_{M,\mathrm{a}} +$ $\mathbf{A}_M\left(\mathbf{x}_{M,\mathrm{a}} - \mathbf{x}_{T,\mathrm{a}}\right)$ | $\hat{c}_T$ | $\hat{c}_{T\leftarrow M'} = \hat{c}_{T,\mathrm{a}} +$ $\sum_j^{70} h_j a_{T,j}\left(\hat{x}'_M - x_{T,\mathrm{a}}\right)_j$ |
| II | $\mathbf{x}_{T,\mathrm{a}}$ | $\hat{\mathbf{x}}_T = \gamma_T \mathbf{x}_{T,\mathrm{a}}$ | $\hat{c}'_M = \hat{c}_M + c_{T,\mathrm{a}} - c_{M,\mathrm{a}} +$ $\sum_j^{10} a_{M,j}\left(\log_{10}x_{M,\mathrm{a}} - \log_{10}x_{T,\mathrm{a}}\right)_j$ | $\hat{c}_{M\leftarrow T} = c_{T,\mathrm{a}} +$ $\sum_j^{10} a_{M,j}\left(\log_{10}\hat{x}_T - \log_{10}x_{T,\mathrm{a}}\right)_j$ |
| III | $\mathbf{x}_{M,\mathrm{a}}$ | $\hat{\mathbf{x}}_M$ | $\hat{c}'_T = \hat{c}_T + c_{M,\mathrm{a}} - c_{T,\mathrm{a}} +$ $\sum_j^{70} h_j a_{T,j}\left(x_{T,\mathrm{a}} - x_{M,\mathrm{a}}\right)_j$ | $\hat{c}_{T\leftarrow M} = c_{M,\mathrm{a}} +$ $\sum_j^{70} h_j a_{T,j}\left(\hat{x}_M - x_{M,\mathrm{a}}\right)_j$ |
| IV | $\mathbf{x}_{M,\mathrm{a}}$ | $\hat{\mathbf{x}}'_T = \frac{\hat{c}'_T}{c_{M,\mathrm{a}}}\mathbf{x}_{M,\mathrm{a}}$ | $\hat{c}_M$ | $\hat{c}_{M\leftarrow T'} = c_{M,\mathrm{a}} +$ $\sum_j^{10} a_{M,j}\left(\log_{10}\hat{x}'_T - \log_{10}x_{M,\mathrm{a}}\right)_j$ |

For method 0, priors, averaging kernels, and surface pressures are not considered. Subscript $T$=TCCON, $M$=MOPITT, a=a priori, the prime symbol represents an adjustment if another prior were used, and the left arrow indicates what would be observed from one system in the absence of error, if the profile from another was taken as truth. The MOPITT column averaging kernel is defined in Eq. 3, the TCCON column averaging kernel is $a_{T,j} = \frac{1}{h_j}\frac{\partial \hat{c}_T}{\partial x_j}$, and the TCCON retrieval scaling factor is $\gamma_T = \frac{\hat{c}_T}{c_{T,\mathrm{a}}}$.

the $\hat{\mathbf{x}}$ profiles may be biased at certain levels. For example, consider the following 2-level example for 3 systems using the same prior where the true state is known:

$$\mathbf{x}_a = \begin{bmatrix} 1 \\ 1 \end{bmatrix},\ \mathbf{x} = \begin{bmatrix} 1 \\ 1.1 \end{bmatrix},\ \mathbf{A}_1 = \begin{bmatrix} 1.1 & 0 \\ 0 & 0.9 \end{bmatrix},\ \mathbf{A}_2 = \begin{bmatrix} 1.2 & 0 \\ 0 & 0.8 \end{bmatrix},\ \mathbf{A}_3 = \begin{bmatrix} 0.9 & 0 \\ 0 & 1.1 \end{bmatrix}. \tag{D1}$$

Then, if $\epsilon_x = 0$ for all three systems

$$\hat{\mathbf{x}}_1 = \begin{bmatrix} 1 \\ 1.09 \end{bmatrix},\ \hat{\mathbf{x}}_2 = \begin{bmatrix} 1 \\ 1.08 \end{bmatrix},\ \hat{\mathbf{x}}_3 = \begin{bmatrix} 1 \\ 1.11 \end{bmatrix},\ \hat{\mathbf{x}}_{12} = \hat{\mathbf{x}}_{21} = \begin{bmatrix} 1 \\ 1.072 \end{bmatrix},\ \hat{\mathbf{x}}_{13} = \hat{\mathbf{x}}_{31} = \begin{bmatrix} 1 \\ 1.099 \end{bmatrix},$$

$$\hat{\mathbf{x}}_{23} = \hat{\mathbf{x}}_{32} = \begin{bmatrix} 1 \\ 1.088 \end{bmatrix},\ \hat{\mathbf{x}}_{11} = \begin{bmatrix} 1 \\ 1.081 \end{bmatrix},\ \hat{\mathbf{x}}_{22} = \begin{bmatrix} 1 \\ 1.064 \end{bmatrix},\ \hat{\mathbf{x}}_{33} = \begin{bmatrix} 1 \\ 1.121 \end{bmatrix}. \tag{D2}$$

where $\hat{\mathbf{x}}_{12}$ is retrieval 2 smoothed with the averaging kernel of 1. In this example $\hat{\mathbf{x}}_1$ is less comparable to $\hat{\mathbf{x}}_{12}$ than to $\hat{\mathbf{x}}_2$, but is more comparable to $\hat{\mathbf{x}}_{13}$ than $\hat{\mathbf{x}}_3$. In all cases, when the system is compared with itself (e.g., $\hat{\mathbf{x}}_1$ and $\hat{\mathbf{x}}_{11}$) the comparison becomes worse. The choice of which profile to apply the AKs to can affect the comparison, and the differences are summarized in Table D2.

In some cases it may be straightforward to make an assumption of a set of true profiles corresponding to soundings to be intercompared. For this project we can readily obtain a set of eight including: $\mathbf{x}_{T,\mathrm{a}}$, $\mathbf{x}_{M,\mathrm{a}}$, $\hat{\mathbf{x}}_T$, $\hat{\mathbf{x}}_M$, $\frac{\hat{c}_M}{c_{T,a}}\mathbf{x}_{T,\mathrm{a}}$, $\frac{\hat{c}_T}{c_{M,a}}\mathbf{x}_{M,\mathrm{a}}$,





**Table D2.** Differences between retrieved and smoothed profiles.

|  | $\mathbf{x}_{i1}$ | $\mathbf{x}_{i2}$ | $\mathbf{x}_{i3}$ |
|---|---|---|---|
| $\mathbf{x}_1 - \mathbf{x}_{1j}$ | 0.009 | 0.018 | −0.009 |
| $\mathbf{x}_2 - \mathbf{x}_{2j}$ | 0.008 | 0.016 | −0.008 |
| $\mathbf{x}_3 - \mathbf{x}_{3j}$ | 0.011 | 0.022 | 0.011 |

Differences between systems (second row) with AKs applied. Rows represent system 1 (or $i$), columns represent system 2 (or $j$). System 2 is smoothed with the AKs of system 1 and compared to system 1 retrievals.

**Table D3.** Simulated differences between retrievals assuming a true profile.

|  | $\mathbf{x}_1$ | $\mathbf{x}_2$ | $\mathbf{x}_3$ | $\mathbf{x}_a$ |
|---|---|---|---|---|
| $\boldsymbol{\epsilon}_1 - \boldsymbol{\epsilon}_2$ | 0.001 | 0.002 | −0.001 | 0.010 |
| $\boldsymbol{\epsilon}_1 - \boldsymbol{\epsilon}_3$ | −0.002 | −0.004 | 0.002 | −0.020 |
| $\boldsymbol{\epsilon}_2 - \boldsymbol{\epsilon}_3$ | −0.003 | −0.006 | 0.003 | −0.030 |

Differences in error between pairs of observing systems (row) using a variety of assumed truth profiles (columns). Differences in error between systems with the same averaging kernels and prior (e.g., $\boldsymbol{\epsilon}_1 - \boldsymbol{\epsilon}_1$) is always zero.

$\frac{\hat{c}_T}{\hat{c}_M}\hat{\mathbf{x}}_M$, and $\frac{\hat{c}_M}{c_{M,a}}\mathbf{x}_{M,\mathrm{a}}$. In the case of 2 profile retrievals:

$$\boldsymbol{\epsilon}_{x,1} - \boldsymbol{\epsilon}_{x,2} = (\hat{\mathbf{x}}_1 - \mathbf{x}_{a,1} + \mathbf{A}_1\mathbf{x}_{a,1} - \hat{\mathbf{x}}_2 + \mathbf{x}_{a,2} - \mathbf{A}_2\mathbf{x}_{a,2}) - \mathbf{A}_1\mathbf{x} + \mathbf{A}_2\mathbf{x} \tag{D3}$$

If the 2 retrievals are not on the same levels then regridding is necessary, for example using a method that conserves layer mass (Appendix C). More specifically for MOPITT and TCCON column retrievals

$$5 \quad \boldsymbol{\epsilon}_{x,M} - \boldsymbol{\epsilon}_{x,T} = \left(\hat{c}_M - c_{a,M} + \mathbf{a}_M^T\log_{10}\mathbf{x}_{a,M} - \hat{c}_T + c_{a,T} - \sum_j^{70} h_j a_{T,j}x_{T,j}\right) - \mathbf{a}_M^T\log_{10}\mathbf{x} + \sum_j^{70} h_j a_{T,j}x_j, \tag{D4}$$

and all the terms relevant to the TCCON retrieval use the TCCON surface pressure, and the terms relevant to the MOPITT retrieval use the MOPITT surface pressure. This way biases from different surface pressures cancel out. The terms in the parenthesis only need to be computed once per pair of compared soundings. For the column retrieval case, regridding is only necessary for the set of profiles treated as truth. Table D3 lists the differences in retrieval errors between systems for a variety
10 of truth profiles. The magnitude of the error depends on the assumption of true profile and may be greater or less than the differences in Table D2.

**Appendix E: Comparison of profiles**

The a priori profiles differ between MOPITT and TCCON. This could lead to different intercomparison results depending on which is chosen as a comparison ensemble. In general, the TCCON a priori profiles are smooth with only 3D variation (time,



latitude, altitude) that takes into account the local tropopause height. MOPITT a priori profiles are 4D (time, latitude, longitude, altitude) and hence differ in locations with strong local pollution (e.g., Pasadena). Examples of prior and retrieved profiles and their differences for several sites are in Fig. E1. Global maps of the average ratios between the retrieved and prior values for MOPITT are available in the SM Sect. S8.

*Author contributions.* JKH and DW were involved in the overall conceptualization, investigation, and methodology development. DW secured funding, computational resources, and provided supervision. TLH wrote the original Sect. 5 draft. JKH did the formal analysis, visualization, and wrote the remainder of the original draft. TLH and DBAJ created methodology for and performed the GEOS-Chem assimilations, and comparisons with HIPPO data. CS provided the AirCore data and MDM, NMD, MKD, DGF, DWTG, FH, LTI, PJ, MK, RK, CL, IM, JN, YSO, HO, DFP, MR, SR, CMR, MS, K Shiomi, K Strong, RS, YT, OU, VAV, WW, TW, POW, and DW provided the TCCON
data which involves independent funding acquisition, site management, data acquisition and processing, QA/QC, and delivery. RBB, and HMW provided guidance on MOPITT data, and insight into the MOPITT instrument, algorithm, and previous validation results. JKH, DW, DBAJ, RRB, NMD, FH, MK, IM, JN, RS, POW, and HMW reviewed the manuscript. JKH and DW implemented edits to the manuscript.

*Competing interests.* We declare that we have no competing interests.

*Acknowledgements.* This project is undertaken with the financial support of the Canadian Space Agency (CSA), through the Earth System
Science Data Analyses program (Grant #16SUASCOBF).

The Ascension Island TCCON station has been supported by the European Space Agency (ESA) under grant 3-14737 and by the German Bundesministerium für Wirtschaft und Energie (BMWi) under grant 50EE1711C. The Four Corners and Manaus TCCON stations have been supported by LANL-LDRD. The Eureka TCCON measurements were made at the Polar Environment Atmospheric Research Laboratory (PEARL) by the Canadian Network for the Detection of Atmospheric Change (CANDAC), primarily supported by the CSA, NSERC, and
Environment and Climate Change Canada (ECCC). The East Trout Lake TCCON station is supported by the Canada Foundation for Innovation, the Ontario Research Fund, and ECCC. Work at Anmyeondo was funded by the Korea Meteorological Administration Research and Development Program "research and development for KMA Wheather, Climate, and Earth system services-support to use of Meteorological Information and Value Crestion" under grant (KMA2018-00321). The TCCON projects for Rikubetsu, Tsukuba, and Burgos sites are supported in part by the GOSAT series project. Site support for Burgos is provided by the Energy Development Corporation (EDC, Philippines).
NMD is supported by an ARC Future Fellowship, FT180100327.

We thank the MOPITT team for providing the MOPITT data - especially Merritt Deeter and James Drummond for helpful discussions. We thank Geoff Toon for developing the GGG2014 code used to process the TCCON data. We acknowledge the TCCON co-investigators and site technicians who have also helped maintain sites and provide data throughout the years, as well as the respective funding organizations that supported the TCCON measurements at the various sites. Specifically we acknowledge Thomas Blumenstock, Youwen Sun, Joseph
Mendonca and Tae-Young Goo.







**Figure E1.** Profiles and profile differences between MOPITT and TCCON for six select sites and different days in 2013. For clarity, only one MOPITT profile within the coincidence criteria is selected per day. The rows are: the TCCON priors ($T_a$), the MOPITT priors ($M_a$), the MOPITT retrieved profiles ($M_r$), the difference between TCCON and MOPITT priors, and the difference between MOPITT priors and retrieved profiles.



We thank Roisin Commane, Enrico Dammers, Benjamin Gaubert, Junjie Liu, Anna Michalak, Charles Miller, Katherine Saad, Mahesh Sha, and Felix Vogel for helpful discussions. We especially thank Merritt Deeter, John Gille, and Geoff Toon for providing feedback on the manuscript.



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
