# Peer review of "Evaluation of MOPITT version 7 joint TIR-NIR XCO retrievals with TCCON"

_Atmospheric Measurement Techniques, 2019_

## Referee Comment (RC1) · Christopher O'Dell (Referee) · 20 Jul 2019

Review of Hedelius et al., 2019. "Evaluation of MOPITT version 7 joint TIR-NIR XCO retrievals with TCCON"

This paper gives a relatively extensive comparison of MOPITT column-average CO (XCO) retrievals with (supposedly more accurate & precise) XCO values from TCCON. The paper is careful to make averaging kernel corrections and correct for a common prior. The paper proposes additional filtering for things like SNR, and proposes a 2-tiered bias correction, an individual correction for each of the four MOPITT pixels, and an over-arching feature-based correction over land, which is related to how "wiggly" the retrieved profile is (the more wiggly, the larger the correction". The paper is chock full of equations, careful tests, and statistical results. It also includes a brief comparison of assimilating the standard MOPITT XCO product into a CO flux inversation system, vs. using the proposed filtering and bias correction.

In general, the paper is well-written. The grammar and spelling are both nearly flawless, for which this reviewer is extremely grateful. However, to some degree this paper is extremely detail-heavy and short on high-level take-aways. The paper could be enhanced by adding some additional high-level conclusions in terms of a) which areas the filters and bias correction help most in, and (b) the real impact of the proposed filters and bias correction in terms of inverse flux modelers at both local scales, as well as regional/global scales. (Ie, do they matter more in the tropics, or high latitudes; more over land vs. ocean, etc). Beyond this limited suggestion, the paper is nearly ready for publication with only minor revisions necessary.

- Sect 2.2 (TCCON), page 6 : Please state the estimated possible mean biases per site (in ppb), and variable errors (noise plus faster systematic errors). You currently only list this single 4% number, which presumably corresponds to the latter.
- Do you consider colocation errors with TCCON, or their possible size? For CO from local emission sources, it seems like colocation errors could be large, depending on your colocation criteria.
- Section 2.3: Please state something about the accuracy and precision of AirCore CO measurements.
- Phase "truth metric" seems an imperfect term. I would prefer truth estimate or truth proxy. "metric" doesn't convey the imperfection of the actual data sources you will use.
- S3, SM, and associated text: What was the source of data you used to construct these plots? Presumably some high-resolution model?
- TCCON-MOPITT trends over the 13 years?

- Figure S4 (filter plot):
    o How can 108.13% of points pass? This problem also exists in S5.

- Why do you show the MODIS snow/ice flag if it passes everything and appears to do nothing?
- SNR, plot c, consider plotting SNR on a log scale so we can see the behavior at low SNR more clearly.
- Plot g, why not also have an upper limit on Tsfc – Tsfc,a ?
- Also in Table 3. Please state what the three tr(S) values are, physically, including units.
- Sum(Retr anom diag) – why is there no histogram?
- Do these filter plots include any bias correction (pixel and/or feature)? Why or why not?

- Figure S5
  - Please fix/add histograms for panels b and d.
  - Given the strong bias and scatter driven by "max diff btwn adj levels", panel j, why not more strongly filter on this variable? Based on the criteria you set in section 3.3, (2 ppb bias, 6 ppb scatter), it seems like a reasonable cut-off would be more like 150 ppb (over land), rather than 300 ppb.

- Page 14, end of section 3: please describe the "maximum difference between adjacent levels" variable physically, and why it might be correlated with bias.
- Section 4.1 - please state if there a noticeable improvement when using the altitude adjustment in comparing to TCCON, in particular for the highest altitude sites, or sites in mountainous regions.
- Page 19, please state at what level you can rule out a long-term trend between MOPITT & TCCON, at for example 95% confidence (2 sigma). 0.1 %/yr ? 0.01 %/yr? Etc.
- Page 20, top: Are any of the observed seasonal biases mitigated using your 1-parameter "feature" bias correction?
- Page 21 bottom to page 22, top: I'm not sure what the DOF and information content analysis is adding. You state numbers but not something that we qualitatively learned. Suggesting removing, or add a sentence or two to true to put the numbers into some kind of context for what we actually learn from them.
- Page 23, top: Because you only assimilate MOPITT, it's not clear what the difference is between assimilations 2 and 3. Are the bias corrections different between the two, i.e. one is still biased high by 6% relative to (bias-corrected) TCCON, and one is not? Please state what the purpose is to have three tests rather than just two. Ie, you are proposing a new filtering and bias correction relative to the standard MOPITT product, so the logical thing to do here would be to test those two things. You are testing three things, so please state clearly why.
- "Assimilation 2 results are lower." Lower in terms of what, inferred CO fluxes? Please clarify.

- Page 24: Finally, in your comparison to HIPPO, it's not clear if any assimilation is clearly "better" in terms of comparison to HIPPO to any other. Please discuss this somewhere in the paper. Currently, your readers cannot tell if your new filtering and bias correction made any real improvement to the MOPITT data in terms of inferred fluxes.
- Appendix D & Section 4.4: In terms of the need to adjust to common priors, and correct for AK differences you never state (1) the general size of these adjustments (in % or ppb), or (2) the effect on the apparent quality of the TCCON vs. MOPITT comparison. Does the comparison improve when you employ one or both of these corretcions? You have gobs of equations and figures but not just a sentence or two saying something like "In general, these corrections are critically important in order to obtain meaningful results" or "In general, these corrections are small and may be in practice ignored", or something in between. Sometimes readers want to skip the equations and numbers and get to the main point!
- Table D3: what are the units? Ppb? % ?

Grammar/Technical:

Page 8: "As a second example, empirical corrections" - please add comma after the leading clause.
Page 9, "To examine the effects of averaging kernels" – please add comma after leading clause.
Page 12: "To reduce the likelihood of overfitting, " – please add comma after leading clause.

Page 13: What is the approximate size of the AK correction typically? Can you apply the AK correction for all MOPITT soundings as compared to TCCON?

---

## Referee Comment (RC2) · Anonymous Referee #2 · 1 Aug 2019

**General Comments**

The manuscript entitled 'Evaluation of MOPITT version 7 joint TIR-NIR X$_{CO}$ retrievals with TCCON' by Hedelius et al. presents results from an intercomparison between MOPITT TIR-NIR data products and CO measurements from the ground-based TCCON network. Covering various technical aspects of intercomparison strategies between the two remote sensing instruments, the manuscript fits well into the scope of AMT. Apart from this detailed validation, it is demonstrated how the different methods of bias correction affect CO emission estimates based on assimilation of MOPITT observations. The paper is well written and the methods are clearly described, with only minor technical and typographical errors. I recommend publication in AMT after minor revisions based on my comments below.

[Figure]

The level of detail and the discussion of various methods applied for the intercomparison studies make the paper quite lengthy. I feel that the readability could be enhanced if parts of these discussions would be moved to the appendix or the supplement. In particular, a large number of correlation plots of MOPITT versus TCCON $X_{CO}$ are presented Section Section 4.2, each of them dealing with the differences in a priori and AKs of both instruments in a different way, with the different methods summarised in Table D1 being difficult to understand. However, there is very little discussion of the impact of these different methods and, in my opinion, not much to learn for the reader. I would therefore suggest to restrict the discussion to the method that has been chosen as standard, and to move Figure 6 to the appendix or supplement.

The manuscript concludes with useful technical suggestions for filtering and bias correction of MOPITT data, and with suggestions for additional information that could enhance future MOPITT data products. What is missing are quantitative statements on the level of agreement between MOPITT and TCCON CO measurements using the different methods presented, an assessment of which method can be recommended, as well as a summary of the lessons learned from the model assimilation studies presented in Section 5. Which of the different assimilations proposed here would be the 'best'?

**Technical Comments**

P2, L21: 'It acts as and indirect greenhouse gas (GHG) as a minor source of $CO_2$...': please check sentence structure

P4, L10: The fact that 'MOPITT data are the longest satellite record of atmospheric CO' has already been stated earlier (P3, L7) and should be removed here.

P5, L23: Insert 'that' between 'They noted' and 'the surface type'

P6, L13: I suggest to remove 'eliminates' with 'is insensitive to' (two times)

P6, L15: 'The network has been around since...' -> 'The network has been established

in...'

P6, L12: The statement that MOPITT data are subject to errors is trivial since this applies to any measurement.

P6, L30: 'long length' -> 'large length'

P8, L18: A first order approximation is always linear (at least if it is based on a Taylor expansion, which I guess is what the authors mean).

P8, L30: Please specify what you mean with 'statistically reasonable products'.

P8, L32: Do you mean $1° \times 1°$ latitude/longitude regions?

Figure 2: It is not clear to me what is shown on the y-axis. The axis label states it would be the difference between retrieved and prior $X_{CO}$, whereas it is stated on P9, L17 that it would be the difference to the weighted mean from all sensors.

P12, L4: I have difficulties to understand this sentence. Do you mean that the bias is determined by subtracting the median from all the $X_{CO}$ measurements within a specific region?

P12, L33: Replace '$\times$' with 'times' (here and anywhere else).

Figure 7: It is not clear which of the various methods to account for a priori and AK differences has been applied to calculate the relative differences between MOPITT and TCCON shown here.

P21, L31: I don't understand what you mean with the statement that the information content is 'a measure how accurate a measurement is to how well a value is a priori known'. $H_s$ expresses the Shannon information content and quantifies to what extent the knowledge about the system is improved by the measurement.

---

## Author Comment (AC1) · 11 Sep 2019

We thank the reviewers for providing feedback on our manuscript. We address their comments below.

**Reviewer comments are in bold-type.**

Our responses are italicized.

We use tracked changes to show changes edits to the manuscript.

Jacob Hedelius (on behalf of all coauthors)

**Reviewer #1**

R1-1: However, to some degree this paper is extremely detail-heavy and short on high-level take-aways. The paper could be enhanced by adding some additional high-level conclusions in terms of a) which areas the filters and bias correction help most in, and (b) the real impact of the proposed filters and bias correction in terms of inverse flux modelers at both local scales, as well as regional/global scales. (Ie, do they matter more in the tropics, or high latitudes; more over land vs. ocean, etc).

We agree the previous conclusion needed to be updated to include take-aways. We have rewritten it and moved the previous text to a new section. Our inversion was too coarse to quantify impact at local scales, but seems to be acting similarly on all regions.

In this study quality filtered and bias corrected MOPITT data are compared with TCCON data. We first derive filters using only the MOPITT data assuming homogeneity over small regions. These filters have the largest affect over snow/ice scenes and over high terrain. They reduce the overall RMS from 3.84 ppb to 2.55 ppb. We find and correct a bias among the four pixels, which we confirmed exists using AirCore. We also find and correct a feature bias.

After the filtering and bias correction, we compare with TCCON data. Using a method (method II, see SM Sect. 6.1) similar to Wunch et al., (2011) to account for differences in priors and AKs we find MOPITT data are biased high by about 6% compared with TCCON, but it is not clear which of MOPITT or TCCON is biased. We also test different methods which all lead to a bias of about 6—10%. There is a trend of -0.06±0.06% yr-1 in the MOPITT-TCCON difference. The bias also appears to depend on site and latitude, but the scatter is not consistent enough to derive a correction. We also compared AKs and information content from the different retrievals. TCCON AKs are more sensitive to changes in the stratosphere. MOPITT AKs peak in the midtroposphere and can vary significantly among locations.

After applying filtering, and an overall scaling to match the TCCON, we assimilate the data into GEOS-Chem. Filtering and bias correction are uniform enough to not make a large difference among regional fluxes. When data are also scaled down to TCCON before implementing into

GEOS-Chem, fluxes were lower in all regions. However, because of bottom-up uncertainties in global CO fluxes these experiments were inconclusive. Additional work is needed to understand the relatively large (~6%) difference between MOPITT and the TCCON.

**R1-2: Sect 2.2 (TCCON), page 6 : Please state the estimated possible mean biases per site (in ppb), and variable errors (noise plus faster systematic errors). You currently only list this single 4% number, which presumably corresponds to the latter.**

Empirical estimates of TCCON XCO site-to-site bias are underway from a campaign in 2018 (Pak et al., 2019 TCCON meeting, New Zealand). The systematic 4% number is also made up of longer-term systematic errors that could vary site-to-site due to e.g., misalignment. GGG2014 XCO systematic errors for TCCON are below 4% (Wunch et al., 2015). The uncertainty on the scaling slope is 6% (20).

**R1-3: Do you consider colocation errors with TCCON, or their possible size? For CO from local emission sources, it seems like colocation errors could be large, depending on your colocation criteria.**

Yes, but only in an overall global bias sense. See former Fig. 6 d-e. In general changing the colocation criteria size (doubling or halving) increased the intercept. While the bias for the base colocation criteria was around 6.4%, halving led to a bias of ~5.9–7.4%, and doubling led to a bias of 5–6.7%. Based on R2-1 we've moved this figure to the supplement but increased discussion in the main text.

We also examine how the scaling changes for different colocation criteria in SM Fig. S12d--e by halving and doubling the coincidence areas. We find MOPITT is biased higher than TCCON in these tests by 5--7.4%. Doubling the area decreases R2 for the global comparison.

**R1-4:** Section 2.3: Please state something about the accuracy and precision of AirCore CO measurements.**

AirCore CO is still a developmental product, so we made the following addition with the caveat that a full uncertainty budget has not yet been quantified.

AirCore CO is still a developmental product with a sample measurement precision typically less than 5 ppb (Engel et al., 2017). However, stratospheric AirCore CO profile comparisons have shown differences as large as 20 ppb, which could be a result of diffusion in stratospheric AirCore samples, AirCore surface effects, or incorrect AirCore sample end member assumptions. Accuracy is dependent on the quality of calibration and standards (see SM Sect. 2).

R1-5: Phase "truth metric" seems an imperfect term. I would prefer truth estimate or truth proxy. "metric" doesn't convey the imperfection of the actual data sources you will use. *Agreed. We've changed to "proxy" throughout.*

**R1-6: S3, SM, and associated text: What was the source of data you used to construct these plots? Presumably some high-resolution model?**

These are generated from the MOPITT data itself. Unfortunately we do not have access to a high-resolution XCO model.

We examine the effects of the bin size on the small region approximation (SRA) using the MOPITT observations.

**R1-7: TCCON-MOPITT trends over the 13 years?**

See response to R1-12.

**R1-8: Figure S4 (filter plot):**

- **R1-8A: How can 108.13% of points pass? This problem also exists in S5.** Before this percentage was compared to SZA and the snow/ice masks determined before the SAA filtering, so in this case the percentage was more than 100 because we were disregarding the snow/ice mask. We've updated the percentages to be compared with all observations up to 90 degrees so they are now under 100%.
- R1-8B: Why do you show the MODIS snow/ice flag if it passes everything and appears to do nothing?

This was because the histogram bars were too narrow to notice, and the distribution is nearly bimodal. We've increased the bin width on the ends to improve visibility.

**• R1-8C: SNR, plot c, consider plotting SNR on a log scale so we can see the behavior at low SNR more clearly.**

This is an excellent idea, we've updated this subplot.

**• R1-8D: Plot g, why not also have an upper limit on Tsfc – Tsfc,a ?**

Limits are all somewhat subjective, and reconsidering it does seem reasonable to have an upper limit here. Because the rest of the analysis was without it we don't include it at this stage, but data users may consider one.

• R1-8E: Also in Table 3. Please state what the three tr(S) values are, physically, including units.

Interpretation physically is more obscure than linear covariance matrices, but is related to uncertainty. We have described these in the table

 $tr(\hat{S})$  unitless due to log scale, related to overall uncert. of retrieval combination of meas & smooth matrices

tr( $\hat{S}_{merr}$ ) from uncert. in prior & weighting tr( $\hat{S}_{smth}$ ) from uncert. in measured radiances

• **R1-8F: Sum(Retr anom diag) – why is there no histogram?** We have increased bin widths to improve visibility. • R1-8G: Do these filter plots include any bias correction (pixel and/or feature)? Why or why not?

They include a pixel BC, but not a feature BC. We've added the following. Mean bias and pixel biases have had a preliminary temporal bias correction applied to bring the values from individual pixels into better agreement. Without this, the truth proxy (local median) would often be a systematically biased value from pixel 1, leading to bimodal distributions for each pixel SRA anomaly. A feature-dependent bias correction was not considered until after generating these plots. The mean bias is only shown for at least 2000 points, and pixel biases are only shown if there are at least 300 points.

**R1-9: Figure S5**

- **R1-9A: Please fix/add histograms for panels b and d.** *Done.*
- R1-9B: Given the strong bias and scatter driven by "max diff btwn adj levels", panel j, why not more strongly filter on this variable? Based on the criteria you set in section 3.3, (2 ppb bias, 6 ppb scatter), it seems like a reasonable cut-off would be more like 150 ppb (over land), rather than 300 ppb.

These are somewhat subjective, and there was internal discussion over this. If readers decided on a 150 ppb cutoff we would not disagree. The hope was that because the relationship looked linear that data could be bias corrected instead of filtered out. In practice the scatter was still large after the bias correction.

**R1-10: Page 14, end of section 3: please describe the "maximum difference between adjacent levels" variable physically, and why it might be correlated with bias.**

We've added the following:

This feature is larger for strong gradients between levels which can appear when there are strong surface fluxes, or when the retrieval is unstable and oscillates. This instability may be caused by bias related to e.g., spectroscopic errors.

**R1-11: Section 4.1 - please state if there a noticeable improvement when using the altitude adjustment in comparing to TCCON, in particular for the highest altitude sites, or sites in mountainous regions.**

In aggregate there is no improvement, as enough high and low values balance out. However, there may be large differences for individual sites.

In this intercomparison, we implicitly account for differences in surface pressure using the **h** vector. -This can make a difference for individual sites by as much as -10.5±4.1 (1 $\sigma$ ) ppb (for Zugspitze). However, we have found in practice that accounting for differences in surface pressure makes little difference here on the overall comparison (compare SM Fig S12c and S12f). In aggregate the difference is only -0.2±1.5 (1 $\sigma$ ) ppb.

**R1-12: Page 19, please state at what level you can rule out a long-term trend between MOPITT & TCCON, at for example 95% confidence (2 sigma). 0.1 %/yr ? 0.01 %/yr? Etc.**

We thank the reviewer for pointing out this oversight. Our previous conclusion of no trend was based only on a visual comparison. When we fit a line despite the scatter, there is a slight negative trend similar to the one found in Deeter et al., 2017.

We do not find a clear long term trend in the MOPITT\$ \$TCCON difference. Though there is significant scatter among individual comparisons, we find a long-term trend of -0.06±0.06% yr-1 in the MOPITT-TCCON difference using the Thiel-Sen estimator.

**R1-13: Page 20, top: Are any of the observed seasonal biases mitigated using your 1-parameter "feature" bias correction?**

There does not appear to be a significant change in seasonal biases from before and after this correction.

R1-14: Page 21 bottom to page 22, top: I'm not sure what the DOF and information content analysis is adding. You state numbers but not something that we qualitatively learned. Suggesting removing, or add a sentence or two to true to put the numbers into some kind of context for what we actually learn from them.

We've added the following:

DOF gives an indication of how many independent parameters can be improved compared with the prior. MOPITT DOFs are between 1 and 2, which indicates total column measurements may be reasonable, but individual layer measurements may not always be accurate.

R1-15: Page 23, top: Because you only assimilate MOPITT, it's not clear what the difference is between assimilations 2 and 3. Are the bias corrections different between the two, i.e. one is still biased high by 6% relative to (bias corrected) TCCON, and one is not? Please state what the purpose is to have three tests rather than just two. Ie, you are proposing a new filtering and bias correction relative to the standard MOPITT product, so the logical thing to do here would be to test those two things. You are testing three things, so please state clearly why. *Yes, assimilation 3 is still biased high relative to the bias corrected TCCON. While it still has the footprint/pixel BC and single "feature" BC, it can be thought of as nearly being not bias corrected compared to TCCON. We've updated the explanation.*

We conducted three experiments in which we assimilated the following datasets: 1) the original MOPITT data, 2) the filtered and bias-corrected data with a scaling down of about 6% to match the standard TCCON data (Fig. 5) (referred to as Assim. 2), 3) the filtered and bias-corrected data with a scaling of less than 0.5% to the TCCON-based data not tied to the WMO scale and without the empirical airmass correction (SM Fig. S12gFig. 6g) (referred to as Assim. 3).

**R1-16: "Assimilation 2 results are lower." Lower in terms of what, inferred CO fluxes? Please clarify.**

*Here, we meant mole fractions. We've modified this sentence.* Assimilation 2 results mole fractions are lower compared to HIPPO than the other assimilations. R1-17: Page 24: Finally, in your comparison to HIPPO, it's not clear if any assimilation is clearly "better" in terms of comparison to HIPPO to any other. Please discuss this somewhere in the paper. Currently, your readers cannot tell if your new filtering and bias correction made any real improvement to the MOPITT data in terms of inferred fluxes.

Unfortunately the results are mixed and hence inconclusive. It's difficult to say which of assimilations are "better" as the comparisons are imperfect. We've added the following paragraph.

These results are inconclusive as to which of the assimilations is best. Comparisons with HIPPO mole fractions are mixed, and uncertainties in the assimilated prior fluxes prevent us from making definitive conclusions from the flux comparison. It is unclear if the filtering and bias corrections improved fluxes in these experiments.

R1-18: Appendix D & Section 4.4: In terms of the need to adjust to common priors, and correct for AK differences you never state (1) the general size of these adjustments (in % or ppb), or (2) the effect on the apparent quality of the TCCON vs. MOPITT comparison. Does the comparison improve when you employ one or both of these corrections? You have gobs of equations and figures but not just a sentence or two saying something like "In general, these corrections are critically important in order to obtain meaningful results" or "In general, these corrections are small and may be in practice ignored", or something in between. Sometimes readers want to skip the equations and numbers and get to the main point! We agree that these sections did not have enough high level conclusions. Per R2-1 comments, we moved Appendix D to the supplement. We have added the following to Section 4.4 We take differences in priors and averaging kernels into account following method II described in the SM Sect. S6.1. Corrections are applied to each MOPITT retrieval and to daily averages of TCCON retrievals within coincidence criteria. We find in practice that corrections change the comparison by about 3%. TCCON data are adjusted by  $0.7\pm1.8$  (10) ppb, and MOPITT data are adjusted by  $-1.0\pm3.1$  (1 $\sigma$ ) ppb.

**R1-19: Table D3: what are the units? Ppb? %?**

Units are ppb or whatever units the state vector is in. We've added to the table title (same for Table D2).

**Grammar/Technical:**

Page 8: "As a second example, empirical corrections" - please add comma after the leading clause.

Page 9, "To examine the effects of averaging kernels" – please add comma after leading clause.

**Page 12: "To reduce the likelihood of overfitting, " – please add comma after leading clause.** *Fixed, thanks.*

**Page 13: What is the approximate size of the AK correction typically? Can you apply the AK correction for all MOPITT soundings as compared to TCCON?**

See response to R1-18. We apply the AK correction to each individual MOPITT sounding.

**Reviewer #2**

R2-1: The level of detail and the discussion of various methods applied for the intercomparison studies make the paper quite lengthy. I feel that the readability could be enhanced if parts of these discussions would be moved to the appendix or the supplement. In particular, a large number of correlation plots of MOPITT versus TCCON XCO are presented in Section 4.2, each of them dealing with the differences in a priori and AKs of both instruments in a different way, with the different methods summarised in Table D1 being difficult to understand. However, there is very little discussion of the impact of these different methods and, in my opinion, not much to learn for the reader. I would therefore suggest to restrict the discussion to the method that has been chosen as standard, and to move Figure 6 to the appendix or supplement.

We agree the paper is quite lengthy, and did have some internal discussion where parts were moved to the supplement. We have moved Figure 6 and Appendix D to the supplement. These were included for completeness – when we presented this work in meetings we were often asked what would happen if we applied the AKs in the opposite way. Each of the methods has its relative merits, and we could simulate situations where they worsen the comparison. We've opted for Method II as standard because it is most similar to what has been used in the past. We have increased discussion on these in the main text.

R2-2A: The manuscript concludes with useful technical suggestions for filtering and bias correction of MOPITT data, and with suggestions for additional information that could enhance future MOPITT data products. What is missing are quantitative statements on the level of agreement between MOPITT and TCCON CO measurements using the different methods presented,

See response to R1-1.

**R2-2B: an assessment of which method can be recommended,**

We now mention method II in the conclusion. This is most similar to a method that has been used in previous studies.

**R2-2C: as well as a summary of the lessons learned from the model assimilation studies presented in Section 5.**

**R2-2D: Which of the different assimilations proposed here would be the 'best'?**

We refrain from stating which of the assimilations, if any, is 'best.' It is difficult to state the best assimilation because the outputs (mixing ratios and fluxes) are themselves compared only with imperfect values.

**Technical Comments**

**P2, L21: 'It acts as an indirect greenhouse gas (GHG) as a minor source of CO2...': please check sentence structure**

*Modified to:* It acts as an indirect greenhouse gas (GHG) as both a minor source of CO2, and by...

**P4, L10: The fact that 'MOPITT data are the longest satellite record of atmospheric CO' has already been stated earlier (P3, L7) and should be removed here. *Fixed, thanks.**

**P5, L23: Insert 'that' between 'They noted' and 'the surface type'** *Fixed.**

**P6, L13: I suggest to remove 'eliminates' with 'is insensitive to' (two times)** *Replaced both occurrences.*

**P6, L15: 'The network has been around since...'** -> **'The network has been established in...'** *Fixed.*

**P6, L12: The statement that MOPITT data are subject to errors is trivial since this applies to any measurement.**

Good point. We've modified to lead sentence. Despite these advantages, there are known sources of uncertainty that could be biasing the measurementsthe data are still subject to errors.

**P6, L30: 'long length' -> 'large length'**

Opted to omit 'long' instead.

**P8, L18: A first order approximation is always linear (at least if it is based on a Taylor expansion, which I guess is what the authors mean).**

Modified to:

To a first order, the biases-Biases in **b** and **c** may be approximated as having a have a linear effect on x (Rodgers, 1990).

**P8, L30: Please specify what you mean with 'statistically reasonable products'.**

Here we meant that while we cannot say much about bias for an individual measurement, in aggregate they may tell us about scatter and bias.

Rather than using proxies that work for each measurement, statistically reasonable products can be used we aggregate many measurements to empirically identify artifacts and outliers.

**P8, L32: Do you mean 1° × 1° latitude/longitude regions?**

Yes, we do mean an area rather than length. Replaced with "~100 km × 100 km"

Figure 2: It is not clear to me what is shown on the y-axis. The axis label states it would be the difference between retrieved and prior XCO, whereas it is stated on P9, L17 that it would be the difference to the weighted mean from all sensors.

Great catch. The y-axis label was incorrect, we've changed to "pxl - mean Xco (ppb)"

**P12, L4: I have difficulties to understand this sentence. Do you mean that the bias is determined by subtracting the median from all the XCO measurements within a specific region?**

Yes.

To calculate anomalies, w $\Psi$ e subtract the median from all the points within that region.

**P12, L33: Replace '×' with 'times' (here and anywhere else).**

*Modified 6 occurrences where '×' was not followed by a number.*

**Figure 7: It is not clear which of the various methods to account for a priori and AK differences has been applied to calculate the relative differences between MOPITT and TCCON shown here.**

Thanks, we've clarified in the caption and in the main text.

Figure 7 shows boxplots of the MOPITT to TCCON differences (using method II) for each site for land only and water only soundings.

Boxplots of the MOPITT-TCCON percent difference at the TCCON sites (using method II), ordered by latitude (degrees north in parenthesis).

**P21, L31: I don't understand what you mean with the statement that the information content is 'a measure how accurate a measurement is to how well a value is a priori known'. Hs expresses the Shannon information content and quantifies to what extent the knowledge about the system is improved by the measurement.**

I do not see a conflict between these descriptions. The extent of knowledge or uncertainty about the system can be described by the covariance matrices. The elements of the retrieved covariance matrix (accuracy of measurement) should be smaller, indicating an improvement in understanding the system, than the a priori covariance (a priori knowledge). In either case, the accompanying equation should make it more clear for readers.